# FakeEdge: Alleviate Dataset Shift in Link Prediction

**Kaiwen Dong**
University of Notre Dame
kdong2@nd.edu

**Yijun Tian**
University of Notre Dame
ytian5@nd.edu

**Zhichun Guo**
University of Notre Dame
zguo5@nd.edu

**Yang Yang**
University of Notre Dame
yyang1@nd.edu

**Nitesh V. Chawla**
University of Notre Dame
nchawla@nd.edu

## Abstract

Link prediction is a crucial problem in graph-structured data. Due to the recent success of graph neural networks (GNNs), a variety of GNN-based models were proposed to tackle the link prediction task. Specifically, GNNs leverage the message passing paradigm to obtain node representation, which relies on link connectivity. However, in a link prediction task, links in the training set are always present while ones in the testing set are not yet formed, resulting in a discrepancy of the connectivity pattern and bias of the learned representation. It leads to a problem of dataset shift which degrades the model performance. In this paper, we first identify the dataset shift problem in the link prediction task and provide theoretical analyses on how existing link prediction methods are vulnerable to it. We then propose FakeEdge, a model-agnostic technique, to address the problem by mitigating the graph topological gap between training and testing sets. Extensive experiments demonstrate the applicability and superiority of FakeEdge on multiple datasets across various domains.

## 1 Introduction

Graph structured data is ubiquitous across a variety of domains, including social networks [1], protein-protein interactions [2], movie recommendations [3], and citation networks [4]. It provides a non-Euclidean structure to describe the relations among entities. The link prediction task is to predict missing links or new forming links in an observed network [5]. Recently, with the success of graph neural networks (GNNs) for graph representation learning [6–9], several GNN-based methods have been developed [10–14] to solve link prediction tasks. These methods encode the representation of target links with the topological structures and node/edge attributes in their local neighborhood. After recognizing the pattern of observed links (training sets), they predict the likelihood of forming new links between node pairs (testing sets) where no link is yet observed.

Nevertheless, existing methods pose a discrepancy of the target link representation between training and testing sets. As the target link is never observed in the testing set by the nature of the task, it will have a different local topological structure when compared to its counterpart from the training set. Thus, the corrupted topological structure shifts the target link representation in the testing set, which we recognize it as a dataset shift problem [15, 16] in link prediction. Note that there are some existing work [11] applying edge masking to moderate such a problem, similar to our treatment. However, they tend to regard it as an empirical trick and fail to identify the fundamental cause as a problem of dataset shift.

We give a concrete example to illustrate how dataset shift can happen in the link prediction task, especially for GNN-based models with message passing paradigm [17] simulating the 1-dimensional Weisfeiler-Lehman (1-WL) test [18]. In Figure 1, we have two local neighborhoods sampled as subgraphs from the training (top) and testing (bottom) set respectively. The node pairs of interest,

K. Dong et al., FakeEdge: Alleviate Dataset Shift in Link Prediction. *Proceedings of the First Learning on Graphs Conference (LoG 2022)*, PMLR 198, Virtual Event, December 9–12, 2022.

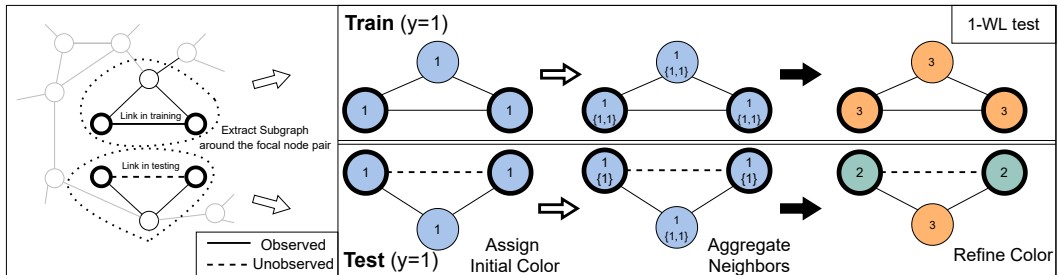

**Figure 1:** 1-WL test is performed to exhibit the learning process of GNNs. Two node pairs (denoted as bold black circles) and their surrounding subgraphs are sampled from the graph as a training (top) and testing (bottom) instance respectively. Two subgraphs are isomorphic when we omit the focal links. One iteration of 1-WL assigns different colors, indicating the occurrence of dataset shift.

which we call focal node pairs, are denoted by black bold circles. From a bird's-eye viewpoint, these two subgraphs are isomorphic when we consider the existence of the positive test link (dashed line), even though the test link has not been observed. Ideally, two isomorphic graphs should have the same representation encoded by GNNs, leading to the same link prediction outcome. However, one iteration of 1-WL in Figure 1 produces different colors for the focal node pairs between training and testing sets, which indicates that the one-layer GNN can encode different representations for these two isomorphic subgraphs, giving rise to dataset shift issue.

Dataset shift can substantially degrade model performance since it violates the common assumption that the joint distribution of inputs and outputs stays the same in both the training and testing set. The root cause of this phenomenon in link prediction is the unique characteristic of the target link: the link always plays a dual role in the problem setting and determines both the input and the output for a link prediction task. The existence of the link apparently decides whether it is a positive or negative sample (output). Simultaneously, the presence of the link can also influence how the representation is learned through the introduction of different topological structures around the link (input). Thus, it entangles representation learning and labels in the link prediction problem.

To decouple the dual role of the link, we advocate a framework, namely **subgraph link prediction**, which disentangles the label of the link and its topological structure. As most practical link prediction methods make a prediction by capturing the local neighborhood of the link [1, 11, 12, 19, 20], we unify them all into this framework, where the input is the extracted subgraph around the focal node pair and the output is the likelihood of forming a link incident with the focal node pair in the subgraph. From the perspective of the framework, we find that the dataset shift issue is mainly caused by the presence/absence of the focal link in the subgraph from the training/testing set. This motivates us to propose a simple but effective technique, **FakeEdge**, to deliberately add or remove the focal link in the subgraph so that the subgraph can stay consistent across training and testing. FakeEdge is a model-agnostic technique, allowing it to be applied to any subgraph link prediction model. It assures that the model would learn the same subgraph representation regardless of the existence of the focal link. Lastly, empirical experiments prove that diminishing the dataset shift issue can significantly boost the link prediction performance on different baseline models.

We summarize our contributions as follows. We first unify most of the link prediction methods into a common framework named as subgraph link prediction, which treats link prediction as a subgraph classification task. In the view of the framework, we theoretically investigate the dataset shift issue in link prediction tasks, which motivates us to propose FakeEdge, a model-agnostic augmentation technique, to ease the distribution gap between the training and testing. We further conduct extensive experiments on a variety of baseline models to reveal the performance improvement with FakeEdge to show its capability of alleviating the dataset shift issue on a broad range of benchmarks.

## 2  Related work

**Dataset Shift.**  Dataset shift is a fundamental issue in the world of machine learning. Within the collection of dataset shift issues, there are several specific problems based on which part of the data experience the distributional shift, including covariate shift, concept shift, and prior probability shift. [16] gives a rigorous definition about different dataset shift situations. In the context of GNNs, [21] investigates the generalization ability of GNN models, and propose a self-supervised task to improve

the size generalization. [22] studies the problem that the node labels in training set are not uniformly sampled and suggests applying a regularizer to reduce the distributional gap between training and testing. [23] proposes a risk minimization method by exploring multiple context of the observed graph to enable GNNs to generalize to out-of-distribution data. [24] demonstrates that the existing link prediction models can fail to generalize to testing set with larger graphs and designs a structural pairwise embedding to achieve size stability. [25–27] study the dataset shift problem for graph-level tasks, especially focusing on the graphs in the training and testing set with varying sizes.

**Graph Data Augmentation.** Several data augmentation methods are introduced to modify the graph connectivity by adding or removing edges [28]. DropEdge [29] acts like a message passing reducer to tackle over-smoothing or overfitting problems [30]. Topping et al. modify the graph's topological structure by removing negatively curved edges to solve the bottleneck issue [32] of message passing [31]. GDC [33] applies graph diffusion methods on the observed graph to generate a diffused counterpart as the computation graph. For the link prediction task, CFLP [34] generates counterfactual links to augment the original graph. Edge Proposal Set [35] injects edges into the training graph, which are recognized by other link predictors in order to improve performance.

## 3 A proposed unified framework for link prediction

In this section, we formally introduce the link prediction task and formulate several existing GNN-based methods into a common general framework.

### 3.1 Preliminary

Let $\mathcal{G} = (V, E, \boldsymbol{x}^V, \boldsymbol{x}^E)$ be an undirected graph. $V$ is the set of nodes with size $n$, which can be indexed as $\{i\}_{i=1}^n$. $E \subseteq V \times V$ is the observed set of edges. $\boldsymbol{x}_i^V \in \mathcal{X}^V$ represents the feature of node $i$. $\boldsymbol{x}_{i,j}^E \in \mathcal{X}^E$ represents the feature of the edge $(i,j)$ if $(i,j) \in E$. The other unobserved set of edges is $E_c \subseteq V \times V \backslash E$, which are either missing or going to form in the future in the original graph $\mathcal{G}$. $d(i,j)$ denotes the shortest path distance between node $i$ and $j$. The $r$-hop *enclosing subgraph* $\mathcal{G}_{i,j}^r$ for node $i, j$ is the subgraph induced from $\mathcal{G}$ by node sets $V_{i,j}^r = \{v | v \in V, d(v,i) \leq r \text{ or } d(v,j) \leq r\}$. The edges set of $\mathcal{G}_{i,j}^r$ are $E_{i,j}^r = \{(p,q)|(p,q) \in E \text{ and } p, q \in V_{i,j}^r\}$. An enclosing subgraph $\mathcal{G}_{i,j}^r = (V_{i,j}^r, E_{i,j}^r, \boldsymbol{x}_{V_{i,j}^r}^V, \boldsymbol{x}_{E_{i,j}^r}^E)$ contains all the information in the neighborhood of node $i, j$. The node set $\{i, j\}$ is called the *focal node pair*, where we are interested in if there exists (observed) or should exist (unobserved) an edge between nodes $i, j$. In the context of link prediction, we will use the term **subgraph** to denote *enclosing subgraph* in the following sections.

### 3.2 Subgraph link prediction

In this section, we discuss the definition of **Subgraph Link Prediction** and investigate how current link prediction methods can be unified in this framework. We mainly focus on link prediction methods based on GNNs, which propagate the message to each node's neighbors in order to learn the representation. We start by giving the definition of the subgraph's properties:

**Definition 1.** *Given a graph $\mathcal{G} = (V, E, \boldsymbol{x}^V, \boldsymbol{x}^E)$ and the unobserved edge set $E_c$, a subgraph $\mathcal{G}_{i,j}^r$ have the following properties:*
*1. a **label** $y \in \{0, 1\}$ of the subgraph indicates if there exists, or will form, an edge incident with focal node pair $\{i, j\}$. That is, $\mathcal{G}_{i,j}^r$ label $y = 1$ if and only if $(i,j) \in E \cup E_c$. Otherwise, label $y = 0$.*
*2. the **existence** $e \in \{0, 1\}$ of an edge in the subgraph indicates whether there is an edge observed at the focal node pair $\{i, j\}$. If $(i,j) \in E$, $e = 1$. Otherwise $e = 0$.*
*3. a **phase** $c \in \{train, test\}$ denotes whether the subgraph belongs to training or testing stage. Especially for a positive subgraph $(y = 1)$, if $(i,j) \in E$, then $c = train$. If $(i,j) \in E_c$, then $c = test$.*

Note that, the *label* $y = 1$ does not necessarily indicate the observation of the edge at the focal node pair $\{i, j\}$. A subgraph in the testing set may have the label $y = 1$ but the edge may not be present. The *existence* $e = 1$ only when the edge is observed at the focal node pair.

**Definition 2.** *Given a subgraph $\mathcal{G}_{i,j}^r$, **Subgraph Link Prediction** is a task to learn a feature $h$ of the subgraph $\mathcal{G}_{i,j}^r$ and uses it to predict the label $y \in \{0, 1\}$ of the subgraph.*

Generally, subgraph link prediction regards the link prediction task as a subgraph classification task. The pipeline of subgraph link prediction starts with extracting the subgraph $\mathcal{G}_{i,j}^r$ around the focal

node pair $\{i, j\}$, and then applies GNNs to encode the node representation $\boldsymbol{Z}$. The latent feature $\mathbf{h}$ of the subgraph is obtained by pooling methods on $\boldsymbol{Z}$. In the end, the subgraph feature $\mathbf{h}$ is fed into a classifier. In summary, the whole pipeline entails:

1. **Subgraph Extraction**: Extract the subgraph $\mathcal{G}_{i,j}^r$ around the focal node pair $\{i, j\}$;
2. **Node Representation Learning**: $\boldsymbol{Z} = \texttt{GNN}(\mathcal{G}_{i,j}^r)$, where $\boldsymbol{Z} \in \mathbb{R}^{|V_{i,j}^r| \times F_{\text{hidden}}}$ is the node embedding matrix learned by the GNN encoder;
3. **Pooling**: $\mathbf{h} = \texttt{Pooling}(\boldsymbol{Z}; \mathcal{G}_{i,j}^r)$, where $\mathbf{h} \in \mathbb{R}^{F_{\text{pooled}}}$ is the latent feature of the subgraph $\mathcal{G}_{i,j}^r$;
4. **Classification**: $\mathbf{y} = \texttt{Classifier}(\mathbf{h})$.

There are two main streams of GNN-based link prediction models. Models like SEAL [11] and WalkPool [12] can naturally fall into the subgraph link prediction framework, as they thoroughly follow the pipeline. In SEAL, SortPooling [36] serves as a readout to aggregate the node's features in the subgraph. WalkPool designs a random-walk based pooling method to extract the subgraph feature $\mathbf{h}$. Both methods take advantage of the node's representation from the entire subgraph.

In addition, there is another stream of link prediction models, such as GAE [10] and PLNLP [14], which learns the node representation and then devises a score function on the representation of the focal node pair to represent the likelihood of forming a link. We find that these GNN-based methods with the message passing paradigm also belong to a subgraph link prediction task. Considering a GAE with $l$ layers, each node $v$ essentially learns its embedding from its $l$-hop neighbors $\{i | i \in V, d(i, v) \leq l\}$. The score function can be then regarded as a center pooling on the subgraph, which only aggregates the features of the focal node pair as $\mathbf{h}$ to represent the subgraph. For a focal node pair $\{i, j\}$ and GAE with $l$ layers, an $l$-hop subgraph $\mathcal{G}_{i,j}^l$ sufficiently contains all the information needed to learn the representation of nodes in the subgraph and score the focal node pair $\{i, j\}$. Thus, the GNN-based models can also be seen as a citizen of subgraph link prediction. In terms of the score function, there are plenty of options depending on the predictive power in practice. In general, the common choices are: (1) Hadamard product: $\mathbf{h} = z_i \circ z_j$; (2) MLP: $\mathbf{h} = \texttt{MLP}(z_i \circ z_j)$ where MLP is the Multi-Layer Perceptron; (3) BiLinear: $\mathbf{h} = z_i \boldsymbol{W} z_j$ where $\boldsymbol{W}$ is a learnable matrix; (4) BiLinearMLP: $\mathbf{h} = \texttt{MLP}(z_i) \circ \texttt{MLP}(z_j)$.

In addition to GNN-based methods, the concept of the subgraph link prediction can be extended to low-order heuristics link predictors, like Common Neighbor [1], Adamic–Adar index [20], Preferential Attachment [37], Jaccard Index [38], and Resource Allocation [39]. The predictors with the order $r$ can be computed by the subgraph $\mathcal{G}_{i,j}^r$. The scalar value can be seen as the latent feature $\mathbf{h}$.

## 4 FakeEdge: Mitigates dataset shift in subgraph link prediction

In this section, we start by giving the definition of dataset shift in the general case, and then formally discuss how dataset shift occurs with regard to subgraph link prediction. Then we propose FakeEdge as a graph augmentation technique to ease the distribution gap of the subgraph representation between the training and testing sets. Lastly, we discuss how FakeEdge can enhance the expressive power of any GNN-based subgraph link prediction model.

### 4.1 Dataset shift

**Definition 3.** *Dataset Shift happens when the joint distribution between train and test is different. That is, $p(\mathbf{h}, \mathbf{y} | \mathbf{c} = \text{train}) \neq p(\mathbf{h}, \mathbf{y} | \mathbf{c} = \text{test})$.*

A simple example of dataset shift is an object detection system. If the system is only designed and trained under good weather conditions, it may fail to capture objects in bad weather. In general, dataset shift is often caused by some unknown latent variable, like the weather condition in the example above. The unknown variable is not observable during the training phase so the model cannot fully capture the conditions during testing. Similarly, the edge existence $\mathbf{e} \in \{0, 1\}$ in the subgraph poses as an "unknown" variable in the subgraph link prediction task. Most of the current GNN-based models neglect the effect of the edge existence on encoding the subgraph's feature.

**Definition 4.** *A subgraph's feature $\mathbf{h}$ is called Edge Invariant if $p(\mathbf{h}, \mathbf{y} | \mathbf{e}) = p(\mathbf{h}, \mathbf{y})$.*

To explain, the **Edge Invariant** subgraph embedding stays the same no matter if the edge is present at the focal node pair or not. It disentangles the edge's existence and the subgraph representation

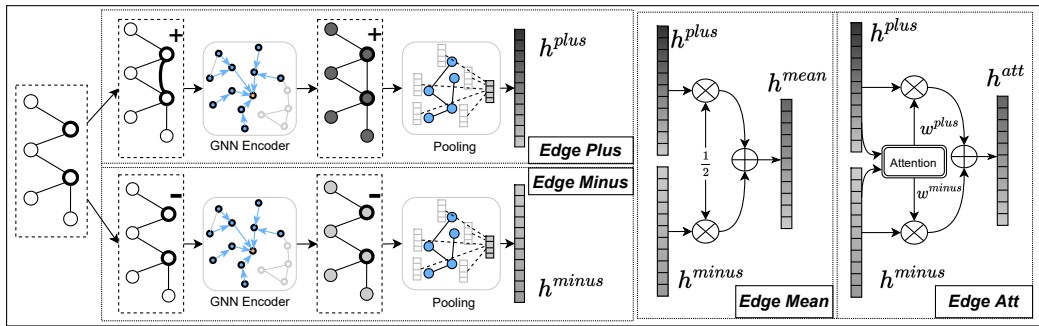

**Figure 2:** The proposed four FakeEdge methods. In general, FakeEdge encourages the link prediction model to learn the subgraph representation by always deliberately adding or removing the edges at the focal node pair in each subgraph. In this way, FakeEdge can reduce the distribution gap of the learned subgraph representation between the training and testing set.

learning. For example, common neighbor predictor is Edge Invariant because the existence of an edge at the focal node pair will not affect the number of common neighbors that two nodes can have. However, Preferential Attachment, another widely used heuristics link prediction predictor, is **not** Edge Invariant because the node degree varies depending on the existence of the edge.

**Theorem 1.** GNN *cannot learn the subgraph feature* **h** *to be Edge Invariant.*

Recall that the subgraphs in Figure 1 are encoded differently between the training and testing set because of the presence/absence of the focal link. Thus, the vanilla GNN cannot learn the Edge Invariant subgraph feature. Learning Edge Invariant subgraph feature is crucial to mitigate the dataset shift problem. Here, we give our main theorem about the issue in the link prediction task:

**Theorem 2.** *Given* $p(\mathbf{h}, \mathbf{y}|\mathbf{e}, \mathbf{c}) = p(\mathbf{h}, \mathbf{y}|\mathbf{e})$*, there is no **Dataset Shift** in the link prediction if the subgraph embedding is **Edge Invariant**. That is,* $p(\mathbf{h}, \mathbf{y}|\mathbf{e}) = p(\mathbf{h}, \mathbf{y}) \implies p(\mathbf{h}, \mathbf{y}|\mathbf{c}) = p(\mathbf{h}, \mathbf{y})$.

The assumption $p(\mathbf{h}, \mathbf{y}|\mathbf{e}, \mathbf{c}) = p(\mathbf{h}, \mathbf{y}|\mathbf{e})$ states that when the edge at the focal node pair is taken into consideration, the joint distribution keeps the same across the training and testing stages, which means that there is no other underlying unobserved latent variable shifting the distribution. The theorem shows an **Edge Invariant** subgraph embedding will not cause a dataset shift phenomenon.

Theorem 2 gives us the motivation to design the subgraph embedding to be Edge Invariant. When it comes to GNNs, the practical GNN is essentially a message passing neural network [17]. The existence of the edge incident at the focal node pair can determine the computational graph for message passing when learning the node representation.

## 4.2 Proposed methods

Having developed conditions of dataset shift phenomenon in link prediction, we next introduce a collection of subgraph augmentation techniques named as **FakeEdge** (Figure 2), which satisfies the conditions in Theorem 2. The motivation is to mitigate the distribution shift of the subgraph embedding by eliminating the different patterns of target link existence between training and testing sets. Note that all of the strategies follow the same discipline: align the topological structure around the focal node pair in the training and testing datasets, especially for the isomorphic subgraphs. Therefore, we expect that it can gain comparable performance improvement across different strategies.

Compared to the vanilla GNN-based subgraph link prediction methods, FakeEdge augments the computation graph for node representation learning and subgraph pooling step to obtain an Edge Invariant embedding for the entire subgraph.

**Edge Plus** A simple strategy is to always make the edge present at the focal node pair for all training and testing samples. Namely, we add an edge into the edge set of subgraph by $E_{i,j}^{r+} = E_{i,j}^r \cup \{(i,j)\}$, and use this edge set to calculate the representation $\mathbf{h}^{plus}$ of the subgraph $\mathcal{G}_{i,j}^{r+}$.

**Edge Minus** Another straightforward modification is to remove the edge at the focal node pair if existing. That is, we remove the edge from the edge set of subgraph by $E_{i,j}^{r-} = E_{i,j}^r \setminus \{(i,j)\}$, and obtain the representation $\mathbf{h}^{minus}$ from $\mathcal{G}_{i,j}^{r-}$.

For GNN-based models, adding or removing edges at the focal node pair can amplify or reduce message propagation along the subgraph. It may also change the connectivity of the subgraph. We are interested to see if it can be beneficial to take both situations into consideration by combining them. Based on *Edge Plus* and *Edge Minus*, we further develop another two Edge Invariant methods:

**Edge Mean**  To combine *Edge Plus* and *Edge Minus*, one can extract these two features and fuse them into one view. One way is to take the average of the two latent features by $\mathbf{h}^{mean} = \frac{\mathbf{h}^{plus} + \mathbf{h}^{minus}}{2}$.

**Edge Att**  *Edge Mean* weighs $\mathcal{G}^{r+}_{i,j}$ and $\mathcal{G}^{r-}_{i,j}$ equally on all subgraphs. To vary the importance of two modified subgraphs, we can apply an adaptive weighted sum operation. Similar to the practice in the text translation [40], we apply an attention mechanism to fuse the $\mathbf{h}^{plus}$ and $\mathbf{h}^{minus}$ by:

$$\mathbf{h}^{att} = w^{plus} * \mathbf{h}^{plus} + w^{minus} * \mathbf{h}^{minus}, \tag{1}$$
$$\text{where } w^{\cdot} = \text{SoftMax}(\boldsymbol{q}^{\mathsf{T}} \cdot tanh(\boldsymbol{W} \cdot \mathbf{h}^{\cdot} + \boldsymbol{b})) \tag{2}$$

### 4.3  Expressive power of structural representation

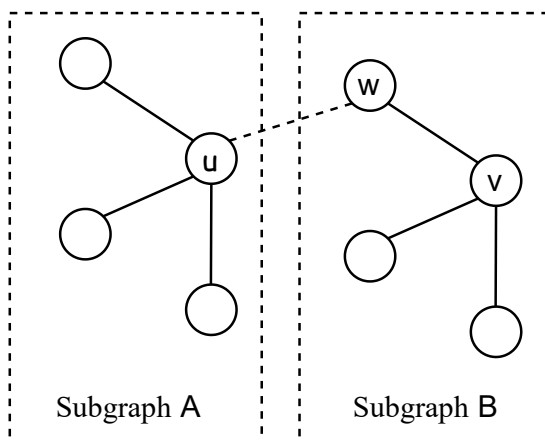

**Figure 3:**  Given two isomorphic but non-overlapping subgraphs $A$ and $B$, GNNs learn the same representation for the nodes $u$ and $v$. Hence, GNN-based methods cannot distinguish focal node pairs $\{u, w\}$ and $\{v, w\}$. However, by adding a FakeEdge at $\{u, w\}$ (shown as the dashed line in the figure), it can break the tie of the representation for $u$ and $v$, thanks to $u$'s modified neighborhood.

In addition to solving the issue of dataset shift, FakeEdge can tackle another problem that impedes the expressive power of link prediction methods on the structural representation [41]. In general, a powerful model is expected to discriminate most of the non-isomorphic focal node pairs. For instance, in Figure 3 we have two isomorphic subgraphs $A$ and $B$, which do not have any overlapping nodes. Suppose that the focal node pairs we are interested in are $\{u, w\}$ and $\{v, w\}$. Obviously, those two focal node pairs have different structural roles in the graph, and we expect different structural representations for them. With GNN-based methods like GAE, the node representation of the node $u$ and $v$ will be the same $z_u = z_v$, due to the fact that they have isomorphic neighborhoods. GAE applies a score function on the focal node pair to pool the subgraph's feature. Hence, the structural representation of node sets $\{u, w\}$ and $\{v, w\}$ would be the same, leaving them inseparable in the embedding space. This issue is caused by the limitation of GNNs, whose expressive power is bounded by 1-WL test [42].

Zhang et al. address this problem by assigning distinct labels between the focal node pair and the rest of the nodes in the subgraph [19]. FakeEdge manages to resolve the issue by augmenting the neighborhoods of those two isomorphic nodes. For instance, we can utilize the *Edge Plus* strategy to deliberately add an edge between nodes $u$ and $w$ (shown as the dashed line in Figure 3). Note that the edge between $v$ and $w$ has already existed. There is no need to add an edge between them. Therefore, the node $u$ and $v$ will have different neighborhoods ($u$ has 4 neighbors and $v$ has 3 neighbors), resulting in the different node representation between the node $u$ and $v$ after the first iteration of message propagation with GNN. In the end, we can obtain different representations for two focal node pairs. Other FakeEdge methods like *Edge Minus* can also tackle the issue in a similar way.

According to Theorem 2 in [19], such non-isomorphic focal node pairs $\{u, w\}$, $\{v, w\}$ are not sporadic cases in a graph. Given an $n$ nodes graph whose node degree is $\mathcal{O}(\log^{\frac{1-\epsilon}{2r}} n)$ for any constant $\epsilon > 0$, there exists $\omega(n^{2\epsilon})$ pairs of such kind of $\{u, w\}$ and $\{v, w\}$, which cannot be distinguished by GNN-based models like GAE. However, FakeEdge can enhance the expressive power of link prediction methods by modifying the subgraph's local connectivity.

**Table 1:** Comparison with and without FakeEdge (AUC). The best results are highlighted in bold.

| Models | FakeEdge | Cora | Citeseer | Pubmed | USAir | NS | PB | Yeast | C.ele | Power | Router | E.coli |
|---|---|---|---|---|---|---|---|---|---|---|---|---|
| GCN | *Original* | $84.92_{\pm1.95}$ | $77.05_{\pm2.18}$ | $81.58_{\pm4.62}$ | $94.07_{\pm1.50}$ | $96.92_{\pm0.73}$ | $93.17_{\pm0.45}$ | $93.76_{\pm0.65}$ | $88.78_{\pm1.85}$ | $76.32_{\pm4.65}$ | $60.72_{\pm5.88}$ | $95.35_{\pm0.36}$ |
| | *Edge Plus* | $91.94_{\pm0.90}$ | $89.54_{\pm1.17}$ | $97.91_{\pm0.14}$ | $97.10_{\pm1.01}$ | $98.03_{\pm0.72}$ | $95.48_{\pm0.42}$ | $97.86_{\pm0.27}$ | $89.65_{\pm1.74}$ | $\mathbf{85.42_{\pm0.91}}$ | $95.96_{\pm0.41}$ | $98.05_{\pm0.30}$ |
| | *Edge Minus* | $92.01_{\pm0.94}$ | $\mathbf{90.29_{\pm0.88}}$ | $97.87_{\pm0.15}$ | $97.16_{\pm0.97}$ | $\mathbf{98.14_{\pm0.66}}$ | $95.50_{\pm0.43}$ | $\mathbf{97.90_{\pm0.29}}$ | $89.47_{\pm1.86}$ | $85.39_{\pm1.08}$ | $96.05_{\pm0.37}$ | $97.97_{\pm0.31}$ |
| | *Edge Mean* | $91.86_{\pm0.76}$ | $89.61_{\pm0.96}$ | $97.94_{\pm0.13}$ | $97.19_{\pm1.00}$ | $98.08_{\pm0.66}$ | $\mathbf{95.52_{\pm0.43}}$ | $97.70_{\pm0.36}$ | $89.62_{\pm1.82}$ | $85.23_{\pm1.00}$ | $\mathbf{96.08_{\pm0.35}}$ | $\mathbf{98.07_{\pm0.27}}$ |
| | *Edge Att* | $\mathbf{92.06_{\pm0.85}}$ | $88.96_{\pm1.05}$ | $\mathbf{97.96_{\pm0.12}}$ | $\mathbf{97.20_{\pm0.69}}$ | $97.96_{\pm0.39}$ | $95.46_{\pm0.45}$ | $97.65_{\pm0.17}$ | $\mathbf{89.76_{\pm2.06}}$ | $85.26_{\pm1.32}$ | $95.90_{\pm0.47}$ | $98.04_{\pm0.16}$ |
| SAGE | *Original* | $89.12_{\pm0.90}$ | $87.76_{\pm0.97}$ | $94.95_{\pm0.44}$ | $96.57_{\pm0.57}$ | $98.11_{\pm0.48}$ | $94.12_{\pm0.45}$ | $97.11_{\pm0.31}$ | $87.62_{\pm1.63}$ | $79.35_{\pm1.66}$ | $88.37_{\pm1.46}$ | $95.70_{\pm0.44}$ |
| | *Edge Plus* | $93.21_{\pm0.82}$ | $90.88_{\pm0.80}$ | $97.91_{\pm0.14}$ | $97.64_{\pm0.73}$ | $\mathbf{98.72_{\pm0.59}}$ | $95.68_{\pm0.39}$ | $98.20_{\pm0.13}$ | $\mathbf{90.94_{\pm1.48}}$ | $86.36_{\pm0.97}$ | $\mathbf{96.46_{\pm0.38}}$ | $98.41_{\pm0.19}$ |
| | *Edge Minus* | $92.45_{\pm0.78}$ | $90.14_{\pm1.04}$ | $97.93_{\pm0.14}$ | $97.50_{\pm0.67}$ | $98.66_{\pm0.55}$ | $95.57_{\pm0.39}$ | $98.13_{\pm0.10}$ | $90.83_{\pm1.59}$ | $85.62_{\pm1.17}$ | $92.91_{\pm1.09}$ | $98.34_{\pm0.26}$ |
| | *Edge Mean* | $92.77_{\pm0.69}$ | $90.60_{\pm0.94}$ | $97.96_{\pm0.13}$ | $\mathbf{97.67_{\pm0.70}}$ | $98.62_{\pm0.61}$ | $\mathbf{95.69_{\pm0.37}}$ | $98.20_{\pm0.13}$ | $90.64_{\pm1.88}$ | $\mathbf{86.46_{\pm0.91}}$ | $96.31_{\pm0.59}$ | $98.41_{\pm0.21}$ |
| | *Edge Att* | $\mathbf{93.31_{\pm1.02}}$ | $\mathbf{91.01_{\pm1.14}}$ | $\mathbf{98.01_{\pm0.13}}$ | $97.40_{\pm0.94}$ | $98.70_{\pm0.59}$ | $95.49_{\pm0.49}$ | $\mathbf{98.22_{\pm0.24}}$ | $90.64_{\pm1.88}$ | $86.46_{\pm0.91}$ | $96.31_{\pm0.59}$ | $\mathbf{98.43_{\pm0.13}}$ |
| GIN | *Original* | $82.70_{\pm1.93}$ | $77.85_{\pm2.64}$ | $91.32_{\pm1.13}$ | $94.89_{\pm0.89}$ | $96.05_{\pm1.10}$ | $92.95_{\pm0.51}$ | $94.50_{\pm0.65}$ | $85.23_{\pm2.56}$ | $73.29_{\pm3.88}$ | $84.29_{\pm1.20}$ | $94.34_{\pm0.57}$ |
| | *Edge Plus* | $90.72_{\pm1.11}$ | $89.54_{\pm1.19}$ | $\mathbf{97.63_{\pm0.14}}$ | $96.03_{\pm1.37}$ | $98.51_{\pm0.55}$ | $95.38_{\pm0.35}$ | $\mathbf{97.84_{\pm0.40}}$ | $\mathbf{89.71_{\pm1.26}}$ | $\mathbf{86.61_{\pm0.87}}$ | $\mathbf{95.79_{\pm0.48}}$ | $97.67_{\pm0.23}$ |
| | *Edge Minus* | $89.88_{\pm1.26}$ | $89.30_{\pm1.08}$ | $97.27_{\pm0.17}$ | $96.36_{\pm0.83}$ | $98.62_{\pm0.45}$ | $95.35_{\pm0.35}$ | $97.80_{\pm0.41}$ | $89.40_{\pm1.91}$ | $86.55_{\pm0.83}$ | $95.72_{\pm0.45}$ | $97.33_{\pm0.36}$ |
| | *Edge Mean* | $90.30_{\pm1.22}$ | $89.47_{\pm1.13}$ | $97.53_{\pm0.19}$ | $\mathbf{96.45_{\pm0.90}}$ | $\mathbf{98.66_{\pm0.45}}$ | $\mathbf{95.39_{\pm0.37}}$ | $97.78_{\pm0.40}$ | $89.66_{\pm2.00}$ | $86.51_{\pm0.92}$ | $95.73_{\pm0.43}$ | $97.57_{\pm0.32}$ |
| | *Edge Att* | $\mathbf{90.76_{\pm0.88}}$ | $\mathbf{89.55_{\pm0.61}}$ | $97.50_{\pm0.15}$ | $96.34_{\pm0.82}$ | $98.35_{\pm0.54}$ | $95.29_{\pm0.29}$ | $97.66_{\pm0.33}$ | $89.39_{\pm1.61}$ | $86.21_{\pm0.67}$ | $95.78_{\pm0.52}$ | $\mathbf{97.74_{\pm0.33}}$ |
| PLNLP | *Original* | $82.37_{\pm1.70}$ | $82.93_{\pm1.73}$ | $87.36_{\pm4.90}$ | $95.37_{\pm0.87}$ | $97.86_{\pm0.93}$ | $92.99_{\pm0.71}$ | $95.09_{\pm1.47}$ | $88.31_{\pm2.21}$ | $81.59_{\pm4.31}$ | $86.41_{\pm1.63}$ | $90.63_{\pm1.68}$ |
| | *Edge Plus* | $91.62_{\pm0.87}$ | $\mathbf{89.88_{\pm1.19}}$ | $98.31_{\pm0.21}$ | $98.09_{\pm0.73}$ | $\mathbf{98.77_{\pm0.39}}$ | $\mathbf{95.33_{\pm0.39}}$ | $98.10_{\pm0.33}$ | $\mathbf{91.77_{\pm2.16}}$ | $90.04_{\pm0.57}$ | $\mathbf{96.45_{\pm0.40}}$ | $\mathbf{98.03_{\pm0.23}}$ |
| | *Edge Minus* | $\mathbf{91.84_{\pm1.42}}$ | $88.99_{\pm1.48}$ | $\mathbf{98.44_{\pm0.14}}$ | $97.92_{\pm0.52}$ | $98.59_{\pm0.44}$ | $95.20_{\pm0.34}$ | $98.01_{\pm0.38}$ | $91.60_{\pm2.23}$ | $89.26_{\pm0.58}$ | $95.01_{\pm0.47}$ | $97.80_{\pm0.16}$ |
| | *Edge Mean* | $91.77_{\pm1.49}$ | $89.45_{\pm1.50}$ | $98.36_{\pm0.14}$ | $\mathbf{98.17_{\pm0.60}}$ | $98.66_{\pm0.56}$ | $95.30_{\pm0.37}$ | $\mathbf{98.10_{\pm0.39}}$ | $91.70_{\pm2.18}$ | $90.05_{\pm0.52}$ | $96.29_{\pm0.47}$ | $98.02_{\pm0.20}$ |
| | *Edge Att* | $91.22_{\pm1.34}$ | $88.75_{\pm1.70}$ | $98.41_{\pm0.17}$ | $98.13_{\pm0.61}$ | $98.70_{\pm0.40}$ | $95.32_{\pm0.38}$ | $98.06_{\pm0.37}$ | $91.72_{\pm2.12}$ | $\mathbf{90.08_{\pm0.54}}$ | $96.40_{\pm0.40}$ | $98.01_{\pm0.18}$ |
| SEAL | *Original* | $90.13_{\pm1.94}$ | $87.59_{\pm1.57}$ | $95.79_{\pm0.78}$ | $97.26_{\pm0.58}$ | $97.44_{\pm1.07}$ | $95.06_{\pm0.46}$ | $96.91_{\pm0.45}$ | $88.75_{\pm1.90}$ | $78.14_{\pm3.14}$ | $92.35_{\pm1.21}$ | $97.33_{\pm0.28}$ |
| | *Edge Plus* | $90.01_{\pm1.95}$ | $89.65_{\pm1.22}$ | $97.30_{\pm0.34}$ | $97.34_{\pm0.59}$ | $98.35_{\pm0.63}$ | $95.35_{\pm0.38}$ | $97.67_{\pm0.32}$ | $89.20_{\pm1.86}$ | $85.25_{\pm0.80}$ | $95.47_{\pm0.58}$ | $97.84_{\pm0.25}$ |
| | *Edge Minus* | $91.04_{\pm1.91}$ | $89.74_{\pm1.16}$ | $97.50_{\pm0.33}$ | $97.27_{\pm0.63}$ | $98.17_{\pm0.74}$ | $\mathbf{95.36_{\pm0.47}}$ | $97.64_{\pm0.30}$ | $89.35_{\pm1.98}$ | $\mathbf{85.30_{\pm0.91}}$ | $\mathbf{95.77_{\pm0.79}}$ | $97.79_{\pm0.30}$ |
| | *Edge Mean* | $90.36_{\pm2.17}$ | $\mathbf{89.87_{\pm1.14}}$ | $\mathbf{97.52_{\pm0.34}}$ | $\mathbf{97.38_{\pm0.68}}$ | $98.23_{\pm0.70}$ | $95.30_{\pm0.34}$ | $97.68_{\pm0.33}$ | $89.19_{\pm1.85}$ | $85.30_{\pm0.87}$ | $95.61_{\pm0.64}$ | $97.83_{\pm0.23}$ |
| | *Edge Att* | $\mathbf{91.08_{\pm1.67}}$ | $89.35_{\pm1.43}$ | $97.26_{\pm0.45}$ | $97.04_{\pm0.79}$ | $\mathbf{98.52_{\pm0.57}}$ | $95.19_{\pm0.43}$ | $\mathbf{97.70_{\pm0.40}}$ | $\mathbf{89.37_{\pm1.40}}$ | $85.24_{\pm1.39}$ | $95.14_{\pm0.62}$ | $\mathbf{97.90_{\pm0.33}}$ |
| WalkPool | *Original* | $\mathbf{92.00_{\pm0.79}}$ | $\mathbf{89.64_{\pm1.01}}$ | $97.70_{\pm0.19}$ | $97.83_{\pm0.97}$ | $99.00_{\pm0.45}$ | $94.53_{\pm0.44}$ | $96.81_{\pm0.92}$ | $93.71_{\pm1.11}$ | $82.43_{\pm3.57}$ | $87.46_{\pm7.45}$ | $95.00_{\pm0.90}$ |
| | *Edge Plus* | $91.96_{\pm0.79}$ | $89.49_{\pm0.96}$ | $98.36_{\pm0.13}$ | $97.97_{\pm0.96}$ | $98.99_{\pm0.58}$ | $\mathbf{95.47_{\pm0.32}}$ | $98.28_{\pm0.24}$ | $93.79_{\pm1.11}$ | $91.24_{\pm0.84}$ | $97.31_{\pm0.26}$ | $98.65_{\pm0.17}$ |
| | *Edge Minus* | $91.97_{\pm0.80}$ | $89.61_{\pm1.04}$ | $\mathbf{98.43_{\pm0.10}}$ | $98.03_{\pm0.95}$ | $99.02_{\pm0.54}$ | $\mathbf{95.47_{\pm0.32}}$ | $\mathbf{98.30_{\pm0.23}}$ | $\mathbf{93.83_{\pm1.13}}$ | $\mathbf{91.28_{\pm0.90}}$ | $\mathbf{97.35_{\pm0.28}}$ | $98.66_{\pm0.17}$ |
| | *Edge Mean* | $91.77_{\pm0.74}$ | $89.55_{\pm1.09}$ | $98.39_{\pm0.11}$ | $98.01_{\pm0.89}$ | $99.02_{\pm0.56}$ | $95.47_{\pm0.29}$ | $98.30_{\pm0.24}$ | $93.70_{\pm1.12}$ | $91.26_{\pm0.81}$ | $97.27_{\pm0.29}$ | $98.65_{\pm0.19}$ |
| | *Edge Att* | $91.98_{\pm0.80}$ | $89.36_{\pm0.74}$ | $98.37_{\pm0.19}$ | $\mathbf{98.12_{\pm0.81}}$ | $\mathbf{99.03_{\pm0.50}}$ | $95.47_{\pm0.27}$ | $98.28_{\pm0.24}$ | $93.63_{\pm1.11}$ | $91.25_{\pm0.60}$ | $97.27_{\pm0.27}$ | $\mathbf{98.70_{\pm0.14}}$ |

## 5 Experiments

In this section, we conduct extensive experiments to evaluate how FakeEdge can mitigate the dataset shift issue on various baseline models in the link prediction task. Then we empirically show the distribution gap of the subgraph representation between the training and testing and discuss how the dataset shift issue can worsen with deeper GNNs. The code for the experiment can be found at https://github.com/Barcavin/FakeEdge.

### 5.1 Experimental setup

**Baseline methods.** We show how FakeEdge techniques can improve the existing link prediction methods, including GAE-like models [10], PLNLP [14], SEAL [11], and WalkPool [12]. To examine the effectiveness of FakeEdge, we compare the model performance with subgraph representation learned on the **original** unmodified subgraph and the FakeEdge augmented ones. For GAE-like models, we apply different GNN encoders, including GCN [9], SAGE [43] and GIN [42]. SEAL and WalkPool have already been implemented in the fashion of the subgraph link prediction. However, a subgraph extraction preprocessing is needed for GAE and PLNLP, since they are not initially implemented as the subgraph link prediction. GCN, SAGE, and PLNLP use a score function to pool the subgraph. GCN and SAGE use the Hadamard product as the score function, while MLP is applied for PLNLP (see Section 3.2 for discussions about the score function). Moreover, GIN applies a subgraph-level pooling strategy, called "mean readout" [42], whose pooling is based on the entire subgraph. Similarly, SEAL and WalkPool also utilize the pooling on the entire subgraph to aggregate the representation. More details about the model implementation can be found in Appendix D.

**Benchmark datasets.** For the experiment, we use 3 datasets with node attributes and 8 without attributes. The graph datasets with node attributes are three citation networks: **Cora** [44], **Citeseer** [45], and **Pubmed** [46]. The graph datasets without node attributes are eight graphs in a variety of domains: **USAir** [47], **NS** [48], **PB** [49], **Yeast** [50], **C.ele** [51], **Power** [51], **Router** [52], and **E.coli** [53]. More details about the benchmark datasets can be found in Appendix E.

**Evaluation protocols.** Following the same experimental setting as of [11, 12], the links are split into 3 parts: 85% for training, 5% for validation, and 10% for testing. The links in validation and testing are unobserved during the training phase. We also implement a universal data pipeline for different methods to eliminate the data perturbation caused by train/test split. We perform 10 random

data splits to reduce the performance disturbance. Area under the curve (AUC) [54] is used as the evaluation metrics and is reported by the epoch with the highest score on the validation set.

## 5.2 Results

**FakeEdge on GAE-like models.** The results of models with (*Edge Plus*, *Edge Minus*, *Edge Mean*, and *Edge Att*) and without (*Original*) FakeEdge are shown in Table 1. We observe that FakeEdge is a vital component for all different methods. With FakeEdge, the link prediction model can obtain a significant performance improvement on all datasets. GAE-like models and PLNLP achieve the most remarkable performance improvement when FakeEdge alleviates the dataset shift issue. FakeEdge boosts them by 2%-11% on different datasets. GCN, SAGE, and PLNLP all have a score function as the pooling methods, which is solely based on the focal node pair. In particular, the focal node pair is incident with the target link, which determines how the message passes around it. Therefore, the most severe dataset shift issues happen at the embedding of the focal node pair during the node representation learning step. FakeEdge is expected to bring a notable improvement to these situations.

**Encoder matters.** In addition, the choice of encoder plays an important role when GAE is deployed on the *Original* subgraph. We can see that SAGE shows the best performance without FakeEdge among these 3 encoders. However, after applying FakeEdge, all GAE-like methods achieve comparable better results regardless of the choice of the encoder. We come to a hypothesis that the plain SAGE itself leverages the idea of FakeEdge to partially mitigate the dataset shift issue. Each node's neighborhood in SAGE is a fixed-size set of nodes, which is uniformly sampled from the full neighborhood set. Thus, when learning the node representation of the focal node pair in the positive training sets, it is possible that one node of the focal node pair is not selected as the neighbor of the other node during the neighborhood sampling stage. In this case, the FakeEdge technique *Edge Minus* is applied to modify such a subgraph.

**FakeEdge on subgraph-based models.** In terms of SEAL and WalkPool, FakeEdge can still robustly enhance the model performance across different datasets. Especially for datasets like Power and Router, FakeEdge increases the AUC by over 10% on both methods. Both methods achieve better results across different datasets, except WalkPool model on datasets Cora and Citeseer. One of the crucial components of WalkPool is the walk-based pooling method, which actually operates on both the *Edge Plus* and *Edge Minus* graphs. Different from FakeEdge technique, WalkPool tackles the dataset shift problem mainly on the subgraph pooling stage. Thus, WalkPool shows similar model performance between the *Original* and FakeEdge augmented graphs. Moreover, SEAL and WalkPool have utilized one of the FakeEdge techniques as a trick in their initial implementations. However, they have failed to explicitly point out the fix of dataset shift issue from such a trick in their papers.

**Different FakeEdge techniques.** When comparing different FakeEdge techniques, *Edge Att* appears to be the most stable, with a slightly better overall performance and a smaller variance. However, there is no significant difference between these techniques. This observation is consistent with our expectation since all FakeEdge techniques follow the same discipline to fix the dataset shift issue.

## 5.3 Further discussions

In this section, we conduct experiments to more thoroughly study why FakeEdge can improve the performance of the link prediction methods. We first give an empirical experiment to show how severe the distribution gap can be between training and testing. Then, we discuss the dataset shift issue with deeper GNNs. Last but not the least, we explore how FakeEdge can even improve the performance of heuristics predictors.

### 5.3.1 Distribution gap between the training and testing

FakeEdge aims to produce Edge Invariant subgraph embedding during the training and testing phases in the link prediction task, especially for those positive samples $p(\mathbf{h}|\mathrm{y} = 1)$. That is, the subgraph representation of the positive samples between the training and testing should be difficult, if at all, to be distinguishable from each other. Formally, we ask whether $p(\mathbf{h}|\mathrm{y} = 1, \mathrm{c} = \mathrm{train}) = p(\mathbf{h}|\mathrm{y} = 1, \mathrm{c} = \mathrm{test})$, by conducting an empirical experiment on the subgraph embedding.

We retrieve the subgraph embedding of the positive samples from both the training and testing stages, and randomly shuffle the embedding. Then we classify whether the sample is from training (c = train)

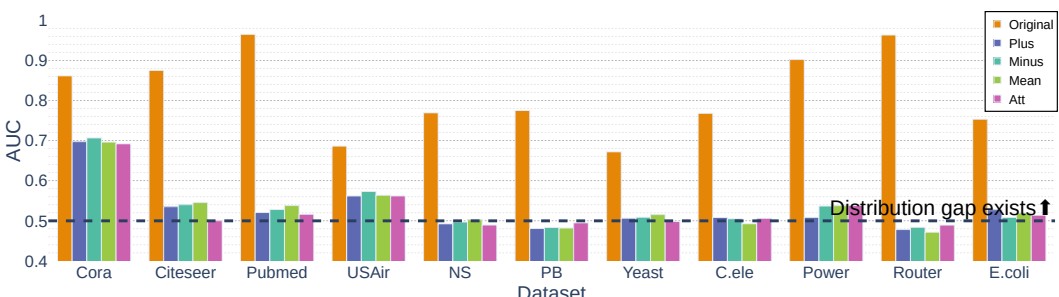

**Figure 4:** Distribution gap (AUC) of the positive samples between the training and testing set.

**Table 2:** GIN's performance improvement by *Edge Att* compared to *Original* with a different number of layers. GIN utilizes mean-pooling as the subgraph-level readout.

| Layers | Cora | Citeseer | Pubmed | USAir | NS | PB | Yeast | C.ele | Power | Router | E.coli |
|--------|------|----------|--------|-------|-----|-----|-------|-------|-------|--------|--------|
| 1 | ↑2.80% | ↑3.65% | ↑4.53% | ↑0.29% | ↑1.30% | ↑1.02% | ↑1.54% | ↑2.13% | ↑5.24% | ↑11.19% | ↑1.67% |
| 2 | ↑4.66% | ↑14.53% | ↑6.64% | ↑0.73% | ↑1.55% | ↑2.16% | ↑3.40% | ↑5.41% | ↑25.32% | ↑14.73% | ↑2.59% |
| 3 | ↑9.78% | ↑15.19% | ↑6.57% | ↑0.98% | ↑2.49% | ↑2.43% | ↑3.60% | ↑4.48% | ↑20.46% | ↑13.38% | ↑3.14% |

or testing (c = test). The shuffled positive samples are split 80%/20% as train and inference sets. Note that the train set here, as well as the inference set, contains both the shuffled positive samples from the training and testing set in the link prediction task. Then we feed the subgraph embedding into a 2-layer MLP classifier to investigate whether the classifier can differentiate the training samples (c = train) and the testing samples (c = test). In general, the classifier will struggle to undertake the classification if the embedding of training and testing samples is drawn from the same underlying distribution, which indicates there is no significant dataset shift issue.

We use GAE with the GCN as the encoder to run the experiment. AUC is used to measure the discriminating power of the classifier. The results are shown in Figure 4. Without FakeEdge, the classifier shows a significant ability to separate positive samples between training and testing. When it comes to the subgraph embedding with FakeEdge, the classifier stumbles in distinguishing the samples. The comparison clearly reveals how different the subgraph embedding can be between the training and testing, while FakeEdge can both provably and empirically diminish the distribution gap.

### 5.3.2 Dataset shift with deeper GNNs

Given two graphs with $n$ nodes in each graph, 1-WL test may take up to $n$ iterations to determine whether two graphs are isomorphic [55]. Thus, GNNs, which mimic 1-WL test, tend to discriminate more non-isomorphic graphs when the number of GNN layers increases. SEAL [19] has empirically witnessed a stronger representation power and obtained more expressive link representation with deeper GNNs. However, we notice that the dataset shift issue in the subgraph link prediction becomes more severe when GNNs try to capture long-range information with more layers.

We reproduce the experiments on GIN by using $l = 1, 2, 3$ message passing layers and compare the model performance by AUC scores with and without FakeEdge. Here we only apply *Edge Att* as the FakeEdge technique. The relative AUC score improvement of *Edge Att* is reported, namely $(AUC_{EdgeAtt} - AUC_{Original})/AUC_{Original}$. The results are shown in Table 2. As we can observe, the relative performance improvement between *Edge Att* and *Original* becomes more significant with more layers, which indicates that the dataset shift issue can be potentially more critical when we seek deeper GNNs for greater predictive power.

To explain such a phenomenon, we hypothesize that GNNs with more layers will involve more nodes in the subgraph, such that their computation graph is dependent on the existence of the edge at the focal node pair. For example, select a node $v$ from the subgraph $\mathcal{G}_{i,j}^r$, which is at least $l$ hops away from the focal node pair $\{i, j\}$, namely $l = min(d(i, v), d(j, v))$. If the GNN has only $l$ layers, $v$ will not include the edge $(i, j)$ in its computation graph. But with a GNN with $l + 1$ layers, the edge $(i, j)$ will affect $v$'s computation graph. We leave the validation of the hypothesis to future work.

**Table 3:** Heuristic methods with/without FakeEdge (AUC). The best results are highlighted in bold.

| Models | Fake Edge | Cora | Citeseer | Pubmed | USAir | NS | PB | Yeast | C.ele | Power | Router | E.coli |
|---|---|---|---|---|---|---|---|---|---|---|---|---|
| PA | *Original* | 63.15±1.38 | 58.20±2.18 | 71.72±0.36 | 88.84±1.41 | 66.19±1.82 | 90.05±0.52 | 82.10±1.15 | 75.72±2.20 | 44.47±1.58 | 48.20±0.83 | 91.99±0.78 |
| | *Edge Plus* | **65.05±1.31** | **61.05±1.96** | **84.04±0.37** | **90.36±1.45** | 65.29±1.97 | **90.47±0.49** | **82.66±0.98** | **75.98±2.31** | 46.83±1.61 | **74.03±1.05** | 91.98±0.78 |
| Jac | *Original* | 71.76±0.85 | 66.33±1.23 | 64.41±0.20 | 88.89±1.55 | 92.19±0.80 | 86.82±0.60 | 88.49±0.53 | 78.77±1.94 | 58.18±0.50 | 55.77±0.55 | 81.43±0.92 |
| | *Edge Plus* | **71.77±0.85** | 66.33±1.23 | **64.42±0.20** | **89.65±1.45** | 92.19±0.80 | **87.20±0.58** | **88.52±0.53** | **79.33±1.88** | 58.18±0.50 | 55.77±0.55 | **81.79±0.90** |

### 5.3.3 Heuristic methods with FakeEdge

FakeEdge, as a model-agnostic technique, not only has the capability of alleviating the dataset shift issue for GNN-based models, but also can tackle the problem for heuristics methods. The heuristics link predictors assign a score to each focal node pair, indicating the likelihood of forming a new edge. Some of the conventional heuristic link predictors, like Common Neighbor [1], Adamic–Adar index [20], or Resource Allocation [39], are Edge Invariant because these predictors are independent of the existence of the target link.

However, other link predictors, including Preferential Attachment (PA) [37] and Jaccard Index (Jac) [38], are not Edge Invariant. The existence/absence of the target link can change the values of the predictors, which in turn changes the ranking of focal node pairs. The original PA for a focal node pair $i, j$ is $PA(i, j) = |\mathcal{N}(i)||\mathcal{N}(j)|$, where $\mathcal{N}(i)$ is the neighbors of node $i$. After applying *Edge Plus*, $PA^{plus}(i, j) = |\mathcal{N}(i) \cup \{j\}||\mathcal{N}(j) \cup \{i\}|$. Similarly, $Jac^{plus}(i, j) = |\mathcal{N}(i) \cap \mathcal{N}(j)|/|\mathcal{N}(i) \cup \mathcal{N}(j) \cup \{i, j\}|$.

We follow the same protocol in the previous experiment. As shown in Table 3, *Edge Plus* can significantly improve the performance of the PA predictor on several datasets. With FakeEdge, PA performs over 10% better on Pubmed. Surprisingly, even though PA is not able to predict the links on Router dataset with AUC score lower than 50%, PA with Edge Plus achieves 74% AUC score and becomes a functional link predictor. In terms of Jac, we observe that Jac with FakeEdge can only gain marginal improvement. This is because that even though Jac is dependent on the existence of target link, the change of Jac index is relatively small when the existence of the target link flips.

## 6 Conclusion

Dataset shift is arguably one of the most challenging problems in the world of machine learning. However, to the best of our knowledge, none of the previous studies sheds light on this notable phenomenon in link prediction. In this paper, we studied the issue of dataset shift in link prediction tasks with GNN-based models. We first unified several existing models into a framework of subgraph link prediction. Then, we theoretically investigated the phenomenon of dataset shift in subgraph link prediction and proposed a model-agnostic technique FakeEdge to amend the issue. Experiments with different models over a wide range of datasets verified the effectiveness of FakeEdge.

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

## A  Related work of link predictions

Early studies on link prediction problems mainly focus on heuristics methods, which require expertise on the underlying trait of network or hand-crafted features, including Common Neighbor [1], Adamic–Adar index [20] and Preferential Attachment [37], etc. WLNM [56] suggests a method to encode the induced subgraph of the target link as an adjacency matrix to represent the link. With the huge success of GNN [9], GNN-based link prediction methods have become dominant across different areas. Graph Auto Encoder(GAE) and Variational Graph Auto Encoder(VGAE) [10] perform link prediction tasks by reconstructing the graph structure. SEAL [11] and DE [13] propose methods to label the nodes according to the distance to the focal node pair. To better exploit the structural motifs [57] in distinct graphs, a walk-based pooling method (WalkPool) [12] is designed to extract the representation of the local neighborhood. PLNLP [14] sheds light on pairwise learning to rank the node pairs of interest. Based on two-dimensional Weisfeiler-Lehman tests, Hu et al. propose a link prediction method that can directly obtain node pair representation [58]. To accelerate the

inference speed, LLP [59] is proposed to perform the link prediction task by distilling the knowledge from GNNs to MLPs.

## B  Proof of Theorem 1

We restate the Theorem 1: GNN cannot learn the subgraph feature **h** to be Edge Invariant.

*Proof.* Recall that the computation of subgraph feature **h** involves steps such as:

1. **Subgraph Extraction**: Extract the subgraph $\mathcal{G}_{i,j}^r$ around the focal node pair $\{i, j\}$;

2. **Node Representation Learning**: $\boldsymbol{Z} = \texttt{GNN}(\mathcal{G}_{i,j}^r)$, where $\boldsymbol{Z} \in \mathbb{R}^{|V_{i,j}^r| \times F_{\text{hidden}}}$ is the node embedding matrix learned by the GNN encoder;

3. **Pooling**: $\mathbf{h} = \texttt{Pooling}(\boldsymbol{Z}; \mathcal{G}_{i,j}^r)$, where $\mathbf{h} \in \mathbb{R}^{F_{\text{pooled}}}$ is the latent feature of the subgraph $\mathcal{G}_{i,j}^r$;

Here, GNN is Message Passing Neural Network [17]. Given a subgraph $\mathcal{G} = (V, E, \boldsymbol{x}^V, \boldsymbol{x}^E)$, GNN with $T$ layers applies following rules to update the representation of node $i \in V$:

$$h_i^{(t+1)} = U_t(h_i^{(t)}, \sum_{w \in \mathcal{N}(i)} M_t(h_i^{(t)}, h_w^{(t)}, \boldsymbol{x}_{i,w}^E)), \tag{3}$$

where $\mathcal{N}(i)$ is the neighborhood of node $i$ in $\mathcal{G}$, $M_t$ is the message passing function at layer $t$ and $U_t$ is the node update function at layer $t$. The hidden states at the first layer are set as $h_i^{(0)} = \boldsymbol{x}_i^V$. The hidden states at the last layer are the outputs $\boldsymbol{Z}_i = h_i^{(T)}$.

Given any subgraph $\mathcal{G}_{i,j}^r = (V_{i,j}^r, E_{i,j}^r, \boldsymbol{x}_{V_{i,j}^r}^V, \boldsymbol{x}_{E_{i,j}^r}^E)$ with the edge present at the focal node pair $(i, j) \in E_{i,j}^r$, we construct another isomorphic subgraph $\mathcal{G}_{\bar{i},\bar{j}}^r = (V_{\bar{i},\bar{j}}^r, E_{\bar{i},\bar{j}}^r, \boldsymbol{x}_{V_{\bar{i},\bar{j}}^r}^V, \boldsymbol{x}_{E_{\bar{i},\bar{j}}^r}^E)$, but remove the edge $(\bar{i}, \bar{j})$ from the edge set $E_{\bar{i},\bar{j}}^r$ of the subgraph. $\mathcal{G}_{\bar{i},\bar{j}}^r$ can be seen as the counterpart of $\mathcal{G}_{i,j}^r$ in the testing set.

Thus, for the first iteration of node updates $t = 1$:

$$h_i^{(1)} = U_t(h_i^{(0)}, \sum_{w \in \mathcal{N}(i)} M_t(h_i^{(0)}, h_w^{(0)}, \boldsymbol{x}_{i,w}^E)), \tag{4}$$

$$h_{\bar{i}}^{(1)} = U_t(h_{\bar{i}}^{(0)}, \sum_{w \in \mathcal{N}(\bar{i})} M_t(h_{\bar{i}}^{(0)}, h_w^{(0)}, \boldsymbol{x}_{\bar{i},w}^E)), \tag{5}$$

Note that $\mathcal{N}(\bar{i}) \cup \{j\} = \mathcal{N}(i)$. We have:

$$h_i^{(1)} = U_t(h_i^{(0)}, \sum_{w \in \mathcal{N}(i) \backslash \{j\}} M_t(h_i^{(0)}, h_w^{(0)}, \boldsymbol{x}_{i,w}^E) + M_t(h_i^{(0)}, h_j^{(0)}, \boldsymbol{x}_{i,j}^E)) \tag{6}$$

$$= U_t(h_{\bar{i}}^{(0)}, \sum_{w \in \mathcal{N}(\bar{i})} M_t(h_{\bar{i}}^{(0)}, h_w^{(0)}, \boldsymbol{x}_{\bar{i},w}^E) + M_t(h_{\bar{i}}^{(0)}, h_{\bar{j}}^{(0)}, \boldsymbol{x}_{\bar{i},\bar{j}}^E)), \tag{7}$$

As $U_t$ is injective, $p(h_i^{(1)}, \mathbf{y} = 1 | \mathbf{e} = 1) \neq p(h_{\bar{i}}^{(1)}, \mathbf{y} = 1) = p(h_i^{(1)}, \mathbf{y} = 1 | \mathbf{e} = 0)$. Similarly, we can conclude that $p(h_i^{(T)}, \mathbf{y} = 1 | \mathbf{e} = 1) \neq p(h_i^{(T)}, \mathbf{y} = 1 | \mathbf{e} = 0)$.

As we use the last iteration of node updates $h_i^{(T)}$ as the final node representation $\boldsymbol{Z}$, we have $p(\boldsymbol{Z}, \mathbf{y} | \mathbf{e} = 1) \neq p(\boldsymbol{Z}, \mathbf{y} | \mathbf{e} = 0)$, which leads to $p(\mathbf{h}, \mathbf{y} | \mathbf{e} = 1) \neq p(\mathbf{h}, \mathbf{y} | \mathbf{e} = 0)$ and concludes the proof. □

## C  Proof of Theorem 2

We restate the Theorem 2: Given $p(\mathbf{h}, \mathbf{y} | \mathbf{e}, \mathbf{c}) = p(\mathbf{h}, \mathbf{y} | \mathbf{e})$, there is no **Dataset Shift** in the link prediction if the subgraph embedding is **Edge Invariant**. That is, $p(\mathbf{h}, \mathbf{y} | \mathbf{e}) = p(\mathbf{h}, \mathbf{y}) \Longrightarrow p(\mathbf{h}, \mathbf{y} | \mathbf{c}) = p(\mathbf{h}, \mathbf{y})$.

*Proof.*

$$p(\mathbf{h} = \boldsymbol{h}, \mathbf{y} = y | \mathbf{c} = c) \tag{8}$$
$$= \mathbb{E}_{\mathbf{e}}[p(\mathbf{h} = \boldsymbol{h}, \mathbf{y} = y | \mathbf{c} = c, \mathbf{e}) p(\mathbf{e} | \mathbf{c} = c)] \tag{9}$$
$$= \mathbb{E}_{\mathbf{e}}[p(\mathbf{h} = \boldsymbol{h}, \mathbf{y} = y) p(\mathbf{e} | \mathbf{c} = c)] \tag{10}$$
$$= p(\mathbf{h} = \boldsymbol{h}, \mathbf{y} = y). \tag{11}$$

$\square$

## D  Details about the baseline methods

To verify the effectiveness of FakeEdge, we tend to introduce minimal modification to the baseline models and make them compatible with FakeEdge techniques. The baseline models in our experiments are mainly from the two streams of link prediction models. One is the GAE-like model, including GCN [9], SAGE [43], GIN [42] and PLNLP [14]. The other includes SEAL [11] and WalkPool [12]. GCN, SAGE and PLNLP learn the node representation and apply a score function on the focal node pair to represent the link. As GAE-like models are not implemented in the fashion of subgraph link prediction, the subgraph extraction step is necessary for them as preprocessing. We follow the code from the labeling trick [19], which implements the GAE models as the subgraph link prediction task. In particular, GIN concatenates the node embedding from different layers to learn the node representation and applies a subgraph-level readout to aggregate as the subgraph representation. As suggested by [13, 19], we always inject the distance information with the Double-Radius Node Labeling [11] (DRNL) to enhance the model performance of GAE-like models. In terms of the selection of hyperparameters, we use the same configuration as [19] on datasets Cora, Citeseer and Pubmed. As they do not have experiments on other 8 networks without attributes, we set the subgraph hop number as 2 and leave the rest of them as default. For PLNLP, we also add a subgraph extraction step without modifying the core part of the pairwise learning strategy. We find that the performance of PLNLP under subgraph setting is very unstable on different train/test splits. In particular, the performance's standard deviation of PLNLP is over $10\%$ on each experiment. Therefore, we also apply DRNL to stabilize the model.

SEAL and WalkPool have applied one of the FakeEdge techniques in their initial implementation. SEAL uses a *Edge Minus* strategy to remove all the edges at focal node pair as a preprocessing step, while WalkPool applies *Edge Plus* to always inject edges into the subgraph for node representation learning. Additionally, WalkPool has the walk-based pooling method operating on both the *Edge Plus* and *Edge Minus* graphs. This design is kept in our experiment. Thus, our FakeEdge technique only takes effect on the node representation step for WalkPool. From the results in Section 5.2, we can conclude that the dataset shift issue on the node representation solely would significantly impact the model performance. We also use the same hyperparameter settings as originally reported in their paper.

## E  Benchmark dataset descriptions

The graph datasets with node attributes are three citation networks: **Cora** [44], **Citeseer** [45] and **Pubmed** [46]. Nodes represent publications and edges represent citation links. The graph datasets without node attributes are: (1) **USAir** [47]: a graph of US Air lines; (2) **NS** [48]: a collaboration network of network science researchers; (3) **PB** [49]: a graph of links between web pages on US political topic; (4) **Yeast** [50]: a protein-protein interaction network in yeast; (5) **C.ele** [51]: the neural network of Caenorhabditis elegans; (6) **Power** [51]: the network of the western USś electric grid; (7) **Router** [52]: the Internet connection at the router-level; (8) **E.coli** [53]: the reaction network of metabolites in Escherichia coli. The detailed statistics of the datasets can be found in Table 4.

## F  Results measured by Hits@20 and statistical significance of results

We adopt another widely used metrics in the link prediction task [60], Hits@20, to evaluate the model performance with and without FakeEdge. The results are shown in Table 5. FakeEdge can boost all the models predictive power on different datasets.

Note that the AUC scores on several datasets are almost saturated in Table 1. To further verify the statistical significance of the improvement, a two-sided $t$-test is conducted with the null hypothesis that

**Table 4:** Statistics of link prediction datasets.

| Dataset | #Nodes | #Edges | Avg. node deg. | Density | Attr. Dimension |
|---|---|---|---|---|---|
| **Cora** | 2708 | 10556 | 3.90 | 0.2880% | 1433 |
| **Citeseer** | 3327 | 9104 | 2.74 | 0.1645% | 3703 |
| **Pubmed** | 19717 | 88648 | 4.50 | 0.0456% | 500 |
| **USAir** | 332 | 4252 | 12.81 | 7.7385% | - |
| **NS** | 1589 | 5484 | 3.45 | 0.4347% | - |
| **PB** | 1222 | 33428 | 27.36 | 4.4808% | - |
| **Yeast** | 2375 | 23386 | 9.85 | 0.8295% | - |
| **C.ele** | 297 | 4296 | 14.46 | 9.7734% | - |
| **Power** | 4941 | 13188 | 2.67 | 0.1081% | - |
| **Router** | 5022 | 12516 | 2.49 | 0.0993% | - |
| **E.coli** | 1805 | 29320 | 16.24 | 1.8009% | - |

**Table 5:** Comparison with and without FakeEdge (Hits@20). The best results are highlighted in bold.

| Models | FakeEdge | Cora | Citeseer | Pubmed | USAir | NS | PB | Yeast | C.ele | Power | Router | E.coli |
|---|---|---|---|---|---|---|---|---|---|---|---|---|
| GCN | *Original* | 65.35±3.64 | 61.71±2.60 | 48.97±1.92 | 87.69±3.92 | 92.77±1.72 | 41.60±2.52 | 85.26±1.90 | 65.33±7.55 | 39.64±5.47 | 39.41±2.38 | 82.21±2.02 |
| | *Edge Plus* | 68.31±2.89 | 65.80±3.28 | **55.70±3.07** | 89.34±4.09 | 93.28±1.69 | 43.98±6.25 | 87.19±2.13 | **66.68±5.25** | 46.92±3.78 | 72.03±2.85 | 86.03±1.40 |
| | *Edge Minus* | 67.97±2.62 | 66.13±3.30 | 54.29±2.66 | 90.57±3.30 | **93.61±1.68** | 43.92±5.82 | 86.66±2.18 | 66.07±6.14 | 47.97±2.58 | **72.34±2.58** | 85.68±1.84 |
| | *Edge Mean* | 67.76±3.02 | 66.11±2.48 | 54.55±2.88 | 89.48±3.52 | 92.77±1.99 | 44.64±6.93 | 86.64±2.03 | 65.28±6.33 | 47.54±2.95 | 72.26±2.68 | 85.62±1.71 |
| | *Edge Att* | **68.43±3.72** | **67.65±4.11** | 55.55±2.70 | **90.80±4.50** | 92.88±2.27 | **44.80±6.60** | **87.83±0.92** | 65.93±11.06 | **48.50±2.20** | 70.96±2.85 | **86.56±1.69** |
| SAGE | *Original* | 61.67±3.68 | 61.10±1.54 | 45.29±3.99 | 89.20±2.80 | 91.93±2.74 | 39.51±4.44 | 84.11±1.47 | 58.55±7.17 | 42.97±5.34 | 30.02±2.75 | 75.30±2.77 |
| | *Edge Plus* | 68.58±2.77 | 65.47±3.58 | **55.23±2.81** | 92.59±3.71 | 93.83±2.54 | **49.10±5.38** | **89.36±0.72** | 69.72±6.02 | **49.70±2.57** | **74.90±3.73** | **88.16±1.29** |
| | *Edge Minus* | 66.26±2.54 | 62.97±3.50 | 53.43±3.52 | 91.32±3.42 | 93.54±1.96 | 48.72±4.90 | 88.27±1.00 | **69.81±5.54** | 47.63±1.87 | 56.67±7.20 | 87.89±1.59 |
| | *Edge Mean* | 66.74±2.71 | 65.96±2.62 | 55.21±2.84 | 91.51±3.49 | 93.25±2.88 | 48.89±6.14 | 89.30±0.72 | 69.21±7.17 | 47.54±3.52 | 73.89±3.50 | 88.05±1.62 |
| | *Edge Att* | **68.80±2.65** | **66.62±3.67** | 55.18±2.99 | **92.92±3.11** | **94.09±1.60** | 48.53±5.15 | 89.10±1.17 | 69.30±7.53 | 47.06±2.21 | 73.60±4.68 | 87.63±1.66 |
| GIN | *Original* | 55.71±4.38 | 51.71±4.31 | 40.14±3.98 | 86.08±3.14 | 90.51±3.45 | 38.79±5.32 | 79.57±1.74 | 54.95±5.91 | 41.56±1.47 | 55.47±4.37 | 77.37±2.84 |
| | *Edge Plus* | **64.42±2.67** | 63.56±2.92 | 49.75±4.50 | 88.68±4.10 | **94.85±1.90** | **46.17±6.12** | 87.58±2.22 | 64.49±6.52 | **48.59±3.33** | 70.67±3.58 | 84.13±2.12 |
| | *Edge Minus* | 63.17±2.96 | 63.65±4.63 | **50.37±4.01** | 89.81±1.80 | 94.53±2.09 | 45.93±6.09 | **88.37±2.00** | **67.06±11.03** | 47.56±1.88 | **71.10±1.90** | 83.23±2.62 |
| | *Edge Mean* | 61.46±4.64 | **63.74±4.20** | 46.97±6.49 | **89.86±2.62** | 93.98±2.88 | 43.48±7.74 | 88.16±2.11 | 66.73±6.79 | 47.66±2.91 | 71.09±2.68 | 82.48±1.99 |
| | *Edge Att* | 63.26±3.33 | 60.64±4.29 | 49.71±4.40 | 88.87±4.71 | 94.49±1.51 | 44.94±5.37 | 87.92±1.45 | 65.93±8.55 | 48.19±2.70 | 70.03±3.05 | **84.38±2.54** |
| PLNLP | *Original* | 58.77±2.59 | 57.21±3.91 | 40.03±3.46 | 88.87±2.75 | 93.76±1.65 | 38.90±4.38 | 81.17±3.54 | 66.36±5.65 | 43.52±6.47 | 34.61±11.29 | 65.68±1.56 |
| | *Edge Plus* | 66.79±2.77 | **67.69±4.13** | 44.44±14.29 | 95.19±1.60 | 95.84±1.09 | 45.18±4.87 | 88.04±2.42 | 71.21±8.04 | **52.37±3.95** | **75.01±1.83** | 84.73±1.70 |
| | *Edge Minus* | 67.40±3.53 | 62.84±2.88 | 47.80±11.11 | 94.10±2.42 | 95.22±1.60 | **45.40±6.29** | 87.94±1.64 | 69.91±6.80 | 52.19±4.23 | 68.24±4.01 | 83.59±1.56 |
| | *Edge Mean* | **68.61±3.40** | 64.81±3.57 | **51.92±13.30** | 95.24±2.09 | **95.95±0.78** | 45.37±5.07 | 88.08±2.30 | **71.26±8.05** | 51.97±3.41 | 74.42±2.33 | 84.78±1.82 |
| | *Edge Att* | 67.82±3.58 | 64.37±3.73 | 48.47±12.01 | **95.38±2.02** | 95.62±0.81 | 45.28±5.11 | **88.57±1.80** | 70.65±8.11 | 51.79±4.07 | 74.99±1.92 | **85.10±1.88** |
| SEAL | *Original* | 60.95±8.00 | 61.56±2.12 | 48.80±3.33 | 91.27±2.53 | 91.72±2.01 | 43.44±6.82 | 85.33±1.76 | 64.21±5.86 | 39.30±3.79 | 59.47±6.66 | 84.15±2.16 |
| | *Edge Plus* | 60.51±7.70 | 65.12±2.18 | 50.90±3.96 | 90.85±4.12 | 93.61±1.87 | 46.77±4.80 | 86.66±1.59 | 65.47±7.68 | 45.90±2.85 | 70.06±3.57 | 85.76±2.04 |
| | *Edge Minus* | 60.74±6.60 | **65.14±2.93** | 51.23±3.82 | 90.66±3.49 | 92.19±2.03 | **47.21±4.73** | 86.49±2.08 | 63.64±6.93 | 46.42±3.42 | **70.43±4.40** | 85.50±2.06 |
| | *Edge Mean* | **62.94±5.78** | 64.99±4.36 | **51.83±3.66** | **91.84±2.93** | 92.92±2.12 | 46.02±4.22 | 86.25±2.17 | **65.93±6.87** | 46.57±3.22 | 70.08±3.85 | 85.85±1.81 |
| | *Edge Att* | 62.03±4.95 | 63.52±4.39 | 48.42±5.69 | 91.42±3.31 | **94.64±1.49** | 44.73±5.29 | **86.83±1.63** | 65.93±4.74 | **47.91±3.45** | 67.46±3.49 | **86.02±1.58** |
| WalkPool | *Original* | 69.98±3.37 | 64.22±2.84 | 57.30±2.56 | 95.09±2.78 | 96.02±1.64 | **47.74±5.81** | 88.24±1.33 | **78.55±5.83** | 43.58±4.40 | 56.21±13.92 | 83.41±1.72 |
| | *Edge Plus* | 69.13±2.31 | **64.51±1.25** | 59.23±3.09 | 95.00±3.09 | 96.06±1.65 | 46.18±5.40 | 89.79±0.70 | 78.36±5.30 | 56.27±4.17 | **77.65±2.83** | 86.44±1.52 |
| | *Edge Minus* | 69.34±2.45 | 64.26±1.93 | 59.44±3.10 | 95.14±2.93 | 95.99±1.67 | 46.79±4.88 | 89.57±0.85 | 77.90±4.49 | 55.72±3.63 | 77.62±2.64 | **87.24±0.77** |
| | *Edge Mean* | **70.27±2.96** | 62.84±4.79 | **59.85±3.84** | 95.24±2.45 | **96.17±1.63** | 46.27±5.00 | 89.58±0.91 | 77.94±4.55 | 56.18±3.74 | 76.88±2.76 | 86.89±0.84 |
| | *Edge Att* | 69.60±4.11 | 64.35±3.64 | 59.63±3.28 | **95.61±2.53** | 96.06±1.62 | 46.77±5.36 | **89.84±0.71** | 77.94±4.89 | **56.46±3.55** | 76.90±2.82 | 87.02±1.64 |

the augmented *Edge Att* and the *Original* representation learning would reach at the identical average scores. The $p$-values of different methods can be found in Table 6. Recall that the $p$-value smaller than 0.05 is considered as statistically significant. GAE-like methods obtain significant improvement on almost all of the datasets, except GCN on C.ele. SEAL shows significant improvement with *Edge Att* on 7 out of 11 datasets. For WalkPool, more than half of the datasets are significantly better.

**Table 6:** $p$-values by comparing AUC scores with *Original* and *Edge Att*. Significant differences are highlighted in bold.

| Models | Cora | Citeseer | Pubmed | USAir | NS | PB | Yeast | C.ele | Power | Router | E.coli |
|---|---|---|---|---|---|---|---|---|---|---|---|
| GCN | $3.50 \cdot 10^{-09}$ | $6.92 \cdot 10^{-12}$ | $1.52 \cdot 10^{-09}$ | $1.10 \cdot 10^{-05}$ | $9.89 \cdot 10^{-04}$ | $1.21 \cdot 10^{-09}$ | $4.95 \cdot 10^{-13}$ | $2.76 \cdot 10^{-01}$ | $1.55 \cdot 10^{-05}$ | $2.62 \cdot 10^{-13}$ | $2.44 \cdot 10^{-14}$ |
| SAGE | $1.32 \cdot 10^{-08}$ | $2.04 \cdot 10^{-06}$ | $3.48 \cdot 10^{-14}$ | $2.78 \cdot 10^{-02}$ | $2.33 \cdot 10^{-02}$ | $4.13 \cdot 10^{-06}$ | $4.87 \cdot 10^{-08}$ | $1.23 \cdot 10^{-03}$ | $6.12 \cdot 10^{-10}$ | $4.40 \cdot 10^{-12}$ | $3.54 \cdot 10^{-13}$ |
| GIN | $4.86 \cdot 10^{-10}$ | $6.09 \cdot 10^{-11}$ | $1.46 \cdot 10^{-12}$ | $1.27 \cdot 10^{-03}$ | $1.29 \cdot 10^{-05}$ | $2.47 \cdot 10^{-10}$ | $5.34 \cdot 10^{-11}$ | $3.84 \cdot 10^{-16}$ | $5.10 \cdot 10^{-09}$ | $3.11 \cdot 10^{-16}$ | $3.04 \cdot 10^{-12}$ |
| PLNLP | $1.47 \cdot 10^{-10}$ | $5.30 \cdot 10^{-07}$ | $1.22 \cdot 10^{-06}$ | $1.66 \cdot 10^{-07}$ | $1.70 \cdot 10^{-02}$ | $3.40 \cdot 10^{-08}$ | $7.69 \cdot 10^{-06}$ | $2.46 \cdot 10^{-03}$ | $7.84 \cdot 10^{-06}$ | $2.68 \cdot 10^{-13}$ | $5.27 \cdot 10^{-11}$ |
| SEAL | $2.59 \cdot 10^{-01}$ | $1.72 \cdot 10^{-02}$ | $6.45 \cdot 10^{-05}$ | $4.82 \cdot 10^{-01}$ | $1.15 \cdot 10^{-02}$ | $5.20 \cdot 10^{-01}$ | $5.91 \cdot 10^{-04}$ | $4.12 \cdot 10^{-01}$ | $3.78 \cdot 10^{-06}$ | $3.91 \cdot 10^{-06}$ | $5.67 \cdot 10^{-04}$ |
| WalkPool | $9.52 \cdot 10^{-01}$ | $4.96 \cdot 10^{-01}$ | $2.83 \cdot 10^{-07}$ | $4.77 \cdot 10^{-01}$ | $8.91 \cdot 10^{-01}$ | $1.84 \cdot 10^{-05}$ | $1.07 \cdot 10^{-04}$ | $8.74 \cdot 10^{-01}$ | $4.15 \cdot 10^{-07}$ | $5.89 \cdot 10^{-04}$ | $1.83 \cdot 10^{-10}$ |

**Table 7:** Model performance with only 20% training data (AUC). The best results are highlighted in bold.

| Models | Fake Edge | Cora | Citeseer | Pubmed | USAir | NS | PB | Yeast | C.ele | Power | Router | E.coli |
|---|---|---|---|---|---|---|---|---|---|---|---|---|
| GCN | *Original* | 55.35±1.07 | 56.02±0.94 | 57.09±1.46 | 80.44±1.44 | 61.14±1.29 | 88.54±0.34 | 81.09±0.53 | 66.67±2.33 | 49.18±0.53 | 62.98±8.98 | 81.79±0.76 |
| | *Edge Plus* | **64.19±0.60** | **62.20±1.07** | **85.65±0.26** | **88.34±0.66** | **62.97±1.56** | 92.40±0.26 | **85.56±0.26** | **71.21±1.26** | **52.77±1.78** | 77.86±0.98 | 91.68±0.27 |
| | *Edge Minus* | 62.89±0.82 | 61.47±0.87 | 85.48±0.28 | 86.46±1.53 | 62.17±1.28 | **92.63±0.26** | 85.44±0.64 | 70.15±1.14 | 52.66±1.26 | 77.68±0.81 | **91.86±0.22** |
| | *Edge Mean* | 63.36±0.75 | 61.76±1.00 | 85.64±0.29 | 87.47±1.18 | 62.91±1.29 | 92.52±0.25 | 84.99±0.47 | 70.84±1.36 | 51.96±1.30 | **77.96±0.88** | 91.81±0.20 |
| | *Edge Att* | 63.30±1.17 | 61.89±0.90 | 85.55±0.29 | 88.19±1.26 | 61.84±1.77 | 92.57±0.24 | 85.51±0.40 | 70.28±1.42 | 52.38±1.53 | 77.44±0.64 | 91.60±0.33 |
| SAGE | *Original* | 51.47±1.68 | 54.02±1.63 | 57.00±7.69 | 84.38±1.14 | 62.54±1.48 | 89.48±0.46 | 77.99±1.05 | 66.06±2.01 | 51.73±0.80 | 61.14±8.58 | 85.01±0.67 |
| | *Edge Plus* | 65.01±0.61 | **63.10±0.63** | **86.90±0.25** | **89.17±0.80** | **65.19±1.75** | **92.71±0.27** | **86.74±0.38** | **72.10±1.81** | **53.99±0.70** | 78.72±0.65 | 91.92±0.15 |
| | *Edge Minus* | 62.71±0.79 | 58.70±1.00 | 85.51±0.21 | 87.24±0.83 | 63.64±1.52 | 91.95±0.30 | 85.61±0.49 | 70.45±1.98 | 52.28±0.68 | 70.66±0.95 | 89.56±0.32 |
| | *Edge Mean* | 64.44±0.82 | 62.63±0.63 | 86.54±0.13 | 88.95±0.72 | 64.33±1.48 | 92.67±0.26 | 86.60±0.30 | 71.78±1.80 | 52.38±0.70 | 78.54±0.70 | 91.86±0.19 |
| | *Edge Att* | **65.31±0.63** | 62.81±1.09 | 86.62±0.24 | 88.73±0.62 | 63.90±1.77 | 92.70±0.22 | 86.64±0.34 | 71.40±1.54 | 53.07±1.32 | **79.08±0.61** | **92.14±0.20** |
| GIN | *Original* | 61.93±0.93 | 61.27±0.95 | 74.32±1.30 | 87.39±0.71 | 62.70±0.81 | 89.52±0.29 | 81.70±0.95 | 68.25±1.65 | 52.02±1.20 | 76.50±0.95 | 90.07±0.51 |
| | *Edge Plus* | **63.30±0.84** | **62.64±1.17** | **86.53±1.08** | 87.63±0.78 | 65.28±1.44 | 91.69±0.35 | 86.30±0.51 | 69.82±1.47 | 53.55±0.92 | 78.08±0.98 | 91.44±0.35 |
| | *Edge Minus* | 62.66±1.08 | 62.11±1.17 | 85.12±0.29 | 87.31±0.85 | 65.01±1.91 | **91.75±0.33** | 86.25±0.58 | 69.72±1.27 | 53.59±0.92 | 78.09±0.93 | 91.39±0.22 |
| | *Edge Mean* | 62.82±1.33 | 62.40±1.10 | 85.67±0.66 | 87.46±0.75 | 65.02±1.65 | 91.72±0.31 | 86.29±0.53 | 69.94±1.53 | 53.54±0.84 | 78.09±1.08 | 91.36±0.26 |
| | *Edge Att* | 63.25±0.70 | 62.07±0.65 | 85.37±0.64 | **87.75±0.92** | **65.54±1.23** | 91.62±0.11 | **86.31±0.44** | **71.47±1.39** | **54.05±0.85** | **78.79±0.66** | **91.49±0.17** |
| PLNLP | *Original* | 63.08±1.01 | 65.23±1.41 | 73.97±1.58 | 84.14±1.94 | 63.06±1.30 | 88.34±1.22 | 81.15±1.52 | 68.33±1.70 | 52.27±0.83 | 68.90±1.08 | 84.80±1.72 |
| | *Edge Plus* | 70.81±0.70 | 72.30±1.59 | **94.57±0.15** | 89.47±0.87 | **65.61±0.80** | **92.66±0.22** | 86.80±0.37 | **73.35±1.44** | **54.19±0.74** | 79.07±0.60 | **92.00±0.29** |
| | *Edge Minus* | 67.21±1.56 | 68.79±1.12 | 93.77±0.13 | 87.49±1.29 | 64.13±1.79 | 92.22±0.24 | 85.78±0.46 | 71.22±1.60 | 52.92±0.70 | 75.97±0.60 | 91.08±0.23 |
| | *Edge Mean* | **72.01±0.96** | **72.79±1.72** | 94.23±0.22 | **89.58±0.89** | 65.33±0.68 | 92.63±0.22 | 86.80±0.43 | 73.07±1.39 | 54.13±0.50 | **80.38±1.10** | 91.93±0.29 |
| | *Edge Att* | 71.62±1.28 | 72.72±0.90 | 94.27±0.22 | 89.49±0.81 | 65.58±0.84 | 92.63±0.21 | **86.80±0.40** | 73.08±1.41 | 54.06±0.54 | 80.37±0.71 | 91.98±0.26 |
| SEAL | *Original* | 58.94±1.72 | 59.12±0.76 | 78.00±1.94 | 87.27±1.30 | 62.59±1.06 | 91.32±0.30 | 83.13±0.83 | 69.25±1.30 | 51.02±0.83 | 71.88±1.43 | 90.89±0.41 |
| | *Edge Plus* | **62.62±1.16** | **62.38±0.91** | 85.31±0.36 | 87.62±0.84 | 63.31±1.41 | 91.67±0.18 | **85.74±0.62** | 69.29±0.97 | 52.86±0.62 | 77.79±0.66 | 91.36±0.43 |
| | *Edge Minus* | 60.75±1.88 | 61.58±0.76 | **86.03±1.31** | 87.54±1.23 | 63.84±0.89 | 91.64±0.21 | 85.43±0.63 | 69.16±1.01 | 52.77±0.86 | **78.06±0.80** | 91.31±0.45 |
| | *Edge Mean* | 61.55±2.03 | 62.19±0.56 | 85.50±0.48 | 87.52±1.00 | 63.27±1.18 | 91.68±0.21 | 85.64±0.56 | **69.30±1.22** | 52.40±0.61 | 77.57±0.90 | **91.41±0.43** |
| | *Edge Att* | 61.77±1.89 | 62.15±0.53 | 85.09±0.53 | **87.76±0.73** | **64.01±1.69** | **91.91±0.36** | 85.56±0.54 | 68.97±1.24 | **53.11±0.95** | 77.82±0.54 | 91.01±0.32 |
| WalkPool | *Original* | 64.05±0.74 | 63.51±1.35 | 86.85±0.83 | 94.54±0.87 | 90.70±0.78 | 93.45±0.36 | 94.26±0.63 | 87.18±0.79 | 65.89±0.76 | 83.63±4.29 | 91.95±1.48 |
| | *Edge Plus* | 64.15±0.87 | 63.49±1.08 | 91.73±0.38 | 94.55±0.92 | 90.57±0.86 | 94.39±0.14 | 94.81±0.24 | **87.22±0.77** | 66.74±0.54 | 87.53±0.40 | 95.30±0.16 |
| | *Edge Minus* | **64.21±0.97** | 63.53±1.04 | **91.81±0.87** | 94.65±0.89 | 90.69±0.86 | 94.39±0.14 | 94.83±0.22 | 87.20±0.76 | 66.77±0.65 | 87.60±0.38 | 95.30±0.15 |
| | *Edge Mean* | 64.13±0.89 | 63.47±1.59 | 91.46±0.86 | 94.65±0.96 | **90.71±0.86** | **94.40±0.13** | 94.83±0.21 | 87.13±0.77 | **66.79±0.61** | 87.57±0.39 | 95.32±0.16 |
| | *Edge Att* | 63.90±1.21 | **63.62±1.39** | 90.97±1.62 | **94.71±0.80** | 90.69±0.81 | 94.40±0.14 | **94.85±0.24** | 87.17±0.82 | 66.78±0.54 | **87.66±0.29** | **95.32±0.13** |

# G   FakeEdge with extremely sparse graphs

In real applications, the size of testing set often outnumbers the training set. When it happens to a link prediction task, the graph will become more sparse because of the huge number of unseen links. We are interested to see how FakeEdge can handle situations where the ratio of training set is low and there exists a lot of "true" links missing in the training graph.

We reset the train/test split as 20% for training, 30% for validation and 50% for testing and reevaluate the model performance. The results can be found in Table 7. As shown in the table, FakeEdge can still consistently improve the model performance under such an extreme setting. It shows that the dataset shift for link prediction is a common issue and FakeEdge has the strength to alleviate it in various settings.

However, we still observe a significant performance drop when compared to the 85/5/10 evaluation setting. This degradation may be caused by a more fundamental dataset shift problem of link prediction: the nodes in a graph are not sampled independently. Existing link prediction models often assume that the likelihood of forming a link relies on its local neighborhood. Nevertheless, an intentionally-sparsified graph can contain a lot of missing links from the testing set, leading to corrupted local neighborhoods of links which cannot reflect the real environments surrounding. FakeEdge does not have the potential to alleviate such a dataset shift. We leave this as a future work.

# H   Concatenation as another valid Edge Invariant subgraph embedding

**Edge Concat**  To fuse the feature from *Edge Plus* and *Edge Minus*, another simple and intuitive way is to concatenate two embedding into one representation. Namely, $\mathbf{h}^{concat} = [\mathbf{h}^{plus}; \mathbf{h}^{minus}]$, where $[\cdot; \cdot]$ is the concatenation operation. $\mathbf{h}^{concat}$ is also an Edge Invariant subgraph embedding. In Table 8, we observe that *Edge Concat* has the similar performance improvement like other FakeEdge methods on all different backbone models.

# I   Dataset shift vs expressiveness: which contributes more with FakeEdge?

In Section 4.3, we discussed how FakeEdge can enhance the expressive power of GNN-based models on non-isomorphic focal node pairs. Meanwhile, we have witnessed the boost of model performance brought by FakeEdge in the experiments. One natural question to ask is whether resolving the dataset shift issue or lifting up expressiveness is the major contributor to make the model perform better.

**Table 8:** Comparison for concatenation operation (AUC). The best results are highlighted in bold.

| Models | Fake Edge | Cora | Citeseer | Pubmed | USAir | NS | PB | Yeast | C.ele | Power | Router | E.coli |
|---|---|---|---|---|---|---|---|---|---|---|---|---|
| GCN | Original | 84.92±1.95 | 77.05±2.18 | 81.58±4.62 | 94.07±1.50 | 96.92±0.73 | 93.17±0.45 | 93.76±0.65 | 88.78±1.85 | 76.32±4.65 | 60.72±5.88 | 95.35±0.36 |
| | Edge Att | 92.06±0.85 | 88.96±1.05 | 97.96±0.12 | 97.20±0.69 | 97.96±0.39 | **95.46±0.45** | **97.65±0.17** | 89.76±2.06 | 85.26±1.32 | 95.90±0.47 | **98.04±0.16** |
| | Edge Concat | **92.63±1.00** | **89.88±1.00** | **97.96±0.11** | **97.27±0.95** | **98.07±0.78** | 95.39±0.44 | 97.55±0.46 | **89.78±1.59** | **85.71±0.75** | **96.19±0.59** | 98.06±0.23 |
| SAGE | Original | 89.12±0.90 | 87.76±0.97 | 94.95±0.44 | 96.57±0.57 | 98.11±0.48 | 94.12±0.45 | 97.11±0.31 | 87.62±1.63 | 79.35±1.66 | 88.37±1.46 | 95.70±0.44 |
| | Edge Att | **93.31±1.02** | 91.01±1.14 | 98.01±0.13 | 97.40±0.94 | **98.70±0.59** | 95.49±0.49 | **98.22±0.24** | 90.64±1.88 | **86.46±0.91** | **96.31±0.59** | **98.43±0.13** |
| | Edge Concat | 93.03±0.57 | **91.14±1.46** | **98.08±0.07** | 97.54±0.70 | 98.59±0.26 | **95.66±0.39** | 98.04±0.37 | **91.14±1.19** | 86.46±1.04 | 96.19±0.54 | 98.40±0.22 |
| GIN | Original | 82.70±1.93 | 77.85±2.64 | 91.32±1.13 | 94.89±0.89 | 96.05±1.10 | 92.95±0.51 | 94.50±0.65 | 85.23±2.56 | 73.29±3.88 | 84.29±1.20 | 94.34±0.57 |
| | Edge Att | 90.76±0.88 | 89.55±0.61 | **97.50±0.15** | **96.34±0.82** | 98.35±0.54 | 95.29±0.29 | 97.66±0.33 | **89.39±1.61** | 86.21±0.67 | **95.78±0.52** | **97.74±0.33** |
| | Edge Concat | **90.90±0.92** | **89.94±0.89** | 97.48±0.16 | 96.17±0.64 | **98.41±0.73** | **95.45±0.39** | **97.71±0.38** | 88.81±1.41 | **86.77±0.99** | 95.72±0.47 | 97.72±0.18 |
| PLNLP | Original | 82.37±1.70 | 82.93±1.73 | 87.36±4.90 | 95.37±0.87 | 97.86±0.93 | 92.99±0.71 | 95.09±1.47 | 88.31±2.21 | 81.59±4.31 | 86.41±1.63 | 90.63±1.68 |
| | Edge Att | 91.22±1.34 | 88.75±1.70 | 98.41±0.17 | **98.13±0.61** | 98.70±0.40 | **95.32±0.38** | **98.06±0.37** | 91.72±2.12 | **90.08±0.54** | **96.40±0.40** | 98.01±0.18 |
| | Edge Concat | **93.01±1.16** | **91.19±1.52** | **98.45±0.12** | 97.86±0.37 | **98.81±0.33** | 95.18±0.24 | 98.04±0.21 | **91.79±1.79** | 89.16±1.01 | 96.31±0.36 | **98.13±0.18** |
| SEAL | Original | 90.13±1.94 | 87.59±1.57 | 95.79±0.78 | **97.26±0.58** | 97.44±1.07 | 95.06±0.46 | 96.91±0.45 | 88.75±1.90 | 78.14±3.14 | 92.35±1.21 | 97.33±0.28 |
| | Edge Att | **91.08±1.67** | 89.35±1.43 | 97.26±0.45 | 97.04±0.79 | **98.52±0.57** | 95.19±0.43 | **97.70±0.40** | **89.37±1.40** | 85.24±1.39 | 95.14±0.62 | **97.90±0.33** |
| | Edge Concat | 90.22±1.60 | **89.93±1.31** | **97.40±0.24** | 96.83±1.01 | 98.23±0.49 | **95.29±0.43** | 97.68±0.34 | 88.99±1.13 | **85.60±1.03** | **95.76±0.74** | 97.72±0.25 |
| WalkPool | Original | **92.00±0.79** | 89.64±1.01 | 97.70±0.19 | 97.83±0.97 | 99.00±0.45 | 94.53±0.44 | 96.81±0.92 | 93.71±1.11 | 82.43±3.57 | 87.46±7.45 | 95.00±0.90 |
| | Edge Att | 91.98±0.80 | 89.36±0.74 | 98.37±0.19 | **98.12±0.81** | 99.03±0.50 | **95.47±0.27** | 98.28±0.24 | 93.63±1.11 | 91.25±0.60 | 97.27±0.27 | **98.70±0.14** |
| | Edge Concat | 91.77±1.06 | **89.79±0.87** | **98.48±0.09** | 98.07±0.86 | **99.05±0.44** | 95.46±0.35 | **98.30±0.25** | **93.82±1.09** | **91.29±0.77** | **97.31±0.27** | 98.70±0.17 |

To answer the question, we first revisit the condition of achieving greater expressiveness. FakeEdge will lift up the expressive power when there exists two nodes being isomorphic in the graph, where we can construct a pair of non-isomorphic focal node pairs which GNNs cannot distinguish. Therefore, how often such isomorphic nodes exist in a graph will determine how much improvement FakeEdge can make by bringing greater expressiveness. Even though isomorphic nodes are common in specific types of graphs like regular graphs, it can be rare in the real world datasets [61]. Thus, we tend to conclude that the effect of solving dataset shift issue by FakeEdge contributes more to the performance improvement rather than greater expressive power. But fully answering the question needs a further rigorous study.

# J  Limitation

FakeEdge can align the embedding of isomorphic subgraphs in training and testing sets. However, it can pose a limitation that hinders one aspect of the GNN-based model's expressive power. Figure 1 gives an example where subgraphs are from training and testing phases, respectively. Now consider that those two subgraphs are both from training set (c = train). Still, the top subgraph has edge observed at focal node pair (y = 1), while the other does not (y = 0). With FakeEdge, two subgraphs will be modified to be isomorphic, yielding the same representation. However, they are non-isomorphic before the modification. To the best of our knowledge, no existing method can simultaneously achieve the most expressive power and get rid of dataset shift issue, because the edge at the focal node pair in the testing set can never be observed under a practical problem setting.

