# OpenReview forum: "FakeEdge: Alleviate Dataset Shift in Link Prediction"
_logconference.io/LOG/2022/Conference — LoG 2022 Poster_

### Official Review · Reviewer_bZPo · 2022-09-29

**Overall Score:** 8
**Confidence:** 3

**Review:**

This work studies link prediction from a novel perspective: the underlying dataset shift issue of applying GNN. It clearly demonstrates such an issue with theory and visualization, and proposes a simple model-agnostic method to fix it. On various datasets and GNNs, the proposed methods lead to significant performance gains.

Strength: this paper is well-written and insightful; the proposed method is simple and effective; and the experiments are rigorous and extensive.

Weakness: I don't see any major weakness in this paper.

Comments:

Although the edge invariance can be achieved in a local subgraph given a focal pair of nodes, the absence/presence of the edge of a focal pair can still be seen by the model when it is involved in the subgraph of another focal pair of nodes. In that case, the model might have a chance to memorize such edge, which could still cause some sort of train/test shift. I'm curious about the authors' thoughts on this argument.

---

### Official Review · Reviewer_Z3rq · 2022-10-21

**Overall Score:** 6
**Confidence:** 4

**Review:**

This paper studies the datasets distribution problem in link prediction, where links in the training set are always present while ones in the testing set are not yet formed. It first shows that exisiting GNN methods cannot mitigate the dataset shift problem. Motivated from this, it further proposes FakeEdge to deliberately add or remove the focal link in the subgraph so that the subgraph can stay consistent across training and testing. Experiments conducted on a variety of datasets demonstrate the effectiveness of the proposed method.

Strong points
	1. The motivation of this paper is interesting and good. Dataset shift problem does exist in link prediction and has longly been overlooked. I think this is meaningful to study the dataset shift problem in link prediction.
	2. The limitations of existing GNNs that cannot handle dataset shift are well-presented and clear.
	3. The proposed method -- FakeEdge is conceptually simple but inspiring.
        4. Experiments are coherent with the motivation, which clearly demonstrates the effectiveness of the proposed method.

Weak points
	1. Just as stated in the paper, there are two mainstream methods for link prediction: 1) GAE and 2) SEAL. GAE methods do suffer from dataset distribution problems as the training edge always exists in the graph, while testing edges do not. However, according to my knowledge, in SEAL, the edge between the focal node pair is always masked for both training and testing edges. So I think the SEAL model can naturally deal with the dataset shift issue (it should be the same as SEAL+Edge_minus). However, in Table 1 we can notice that SEAL does not perform the same as SEAL+Edge_minus. Can you explain the difference between SEAL and SEAL+Edge_minus, if there is any? Otherwise, it will weaken this paper’s contribution.

Generally, I think this is a good paper. The studied problem is interesting, important, and fundamental. The proposed method is simple yet effective. I’d like to give a weak accept.  However, I hope the authors can answer my question above in the rebuttal. Otherwise, I may reduce my rating.

---

### Official Review · Reviewer_radF · 2022-10-22

**Overall Score:** 8
**Confidence:** 4

**Review:**

This paper studies the dataset shift problem in link prediction. The key intuition is that the existence prior knowledge on the existence of edge between a focal node pair could skew the joint distribution of learned representation and the label (i.e., edge existence). The authors theoretically justify that existing GNNs could fail in this scenario because the learned representations are not edge invariant. To mitigate the issue, a family of subgraph augmentation techniques called FakeEdge is proposed, which helps to learn the edge invariant representation of an enclosing subgraph. Experimental evaluations demonstrate that FakeEdge is helpful in improving the link prediction performance and mitigating the discrepancy between the training set and test set.

Strengths:
- There is an increasing discussion on the distributional shift between the training set and test set, so this paper studies an emerging problem in the context of link prediction with GNN.
- The proposed method is simple and intuitive in addressing the dataset shift problem.
- Experimental evaluations support the claim(s) made by the authors.

Concerns:
- Dataset shift in this paper can be formulated as the distributional shift in the training set and test set, which is also closely related to out-of-distribution (OOD) problem (e.g., covariate shift, concept shift). (1) It would be great is the authors could provide more discussion on the similarity/difference between dataset shift and OOD/covariate shift/concept shift; (2) some discussion on the related work of OOD in the context of GNNs might be needed.
- Following my concern above, if these two problems are indeed related, would existing works on OOD+GNNs be helpful in addressing the dataset shift problem as well? I'm interested in some empirical comparison between FakeEdge with this type of methods. If OOD+GNNs is helpful, what are the pros/cons of FakeEdge against this type of methods?
- FakeEdge considers fusion the edge plus and edge minus with either mean pooling or weighted aggregation. What is the benefit of pooling over concatenantion, i.e., h = (h_plus || h_minus) where || is the concatenation operator?
- Section 4.3 states that Edge Plus is more expressive in discriminating the non-isomorphic focal node pairs. However, it seems Edge Plus is less effective than Edge Minus in most cases of the experiments. Could the authors provide some insightful analysis/explanations on that?
- In the experiments, the ratio of training set is pretty high (80%). I am curious what would happen if we set the ratio to a lower number. In addition, what if the ratio of training set is extremely low? This could make the graph to be very sparse, and most of the nodes would be isolated. Would FakeEdge be effective for these isolated nodes in this case?

Minor points:
- [line 147] node pair i, j -> node pair {i, j}
- [line 197] it might be better to say 'a family of subgraph augmentation techniques named FakeEdge' or 'a collection of subgraph augmentation techniques named FakeEdge' rather than 'several straightforward subgraph augmentation techniques, FakeEdge (Figure 2), xxx'. It reads a bit weird and gives an impression that FakeEdge is only one of the augmentation strategy you proposed.

---

### Official Review · Reviewer_5VXk · 2022-10-27

**Overall Score:** 6
**Confidence:** 3

**Review:**

The authors propose FakeEdge to bridge the gap between training data distribution and test data distribution for the link prediction task (dataset shift phenomenon in link prediction).


FakeEdge is a model-agnostic augmentation technique that adds or removes the focal link in the subgraph deliberately to ensure that the subgraph maintains its consistency during training and testing.


FakeEdge augments the computation graph for node representation learning and subgraph pooling phase to achieve an Edge Invariant embedding for the whole subgraph, as opposed to the standard GNN-based subgraph link prediction techniques.


Through a series of experiments in which various models were utilized over several datasets, the usefulness of FakeEdge was demonstrated.

In the field of machine learning, dataset shift is often considered to be one of the most difficult issues to solve.

As a researcher who works in the same field, I am aware that this issue has not yet been addressed, particularly with regard to link prediction.

I think this research can give us some meaningful insight to rethink underlying data distribution for the link prediction task.

---

### Meta-Review · Area_Chair_FN4C · 2022-11-13

**Confidence:** 5
**Recommendation:** Accept

**Meta Review:**

This work addresses the important problem of dataset shift between training and test time that is encountered in practice by many link prediction methods. The proposed solution, FakeEdge, adds or removes focal links in subgraphs such that the training and test distributions are consistent.

The reviewers were unanimously positive about this paper, which addresses an important and understudied problem. The proposed method, while simple, is supported by well-executed empirical results that show significant improvement across a broad range of datasets and is generally applicable to many link prediction methods.

---

### Decision · Program_Chairs · 2022-11-22

Accept (Poster)